# Offline impact of transcranial focused ultrasound on cortical activation in primates

Lennart Verhagen[1,2][†]*, Cécile Gallea[3][†], Davide Folloni[1,2], Charlotte Constans[4], Daria EA Jensen[1,2], Harry Ahnine[3], Léa Roumazeilles[1,2], Mathieu Santin[3], Bashir Ahmed[5], Stéphane Lehericy[3], Miriam C Klein-Flügge[1,2], Kristine Krug[5], Rogier B Mars[2,6], Matthew FS Rushworth[1,2][‡], Pierre Pouget[7][‡], Jean-François Aubry[8][‡], Jerome Sallet[1,2][‡]

[1]Wellcome Centre for Integrative Neuroimaging (WIN), Department of Experimental Psychology, University of Oxford, Oxford, United Kingdom; [2]Wellcome Centre for Integrative Neuroimaging (WIN), Centre for Functional MRI of the Brain (FMRIB), Nuffield Department of Clinical Neurosciences, John Radcliffe Hospital, University of Oxford, Oxford, United Kingdom; [3]Institute du Cerveau et de la Moelle épinière (ICM), Centre for NeuroImaging Research (CENIR), Inserm U 1127, CNRS UMR 7225, Sorbonne Université, Paris, France; [4]Physics for Medicine Paris, Inserm, ESPCI Paris, CNRS, PSL Research University, Université Paris Diderot, Sorbonne Paris Cité, Paris, France; [5]Department of Physiology, Anatomy and Genetics, University of Oxford, Oxford, United Kingdom; [6]Donders Institute for Brain, Cognition and Behaviour, Radboud University Nijmegen, Nijmegen, The Netherlands; [7]Institute du Cerveau et de la Moelle épinière (ICM), UMRS 975 INSERM, CNRS 7225, UMPC, Paris, France; [8]Physics for Medicine Paris, Inserm, ESPCI Paris, CNRS, PSL Research University, Paris, France

*For correspondence:
lennart.verhagen@psy.ox.ac.uk

[†]These authors contributed equally to this work
[‡]These authors also contributed equally to this work

Competing interests: The authors declare that no competing interests exist.

**Abstract** To understand brain circuits it is necessary both to record and manipulate their activity. Transcranial ultrasound stimulation (TUS) is a promising non-invasive brain stimulation technique. To date, investigations report short-lived neuromodulatory effects, but to deliver on its full potential for research and therapy, ultrasound protocols are required that induce longer-lasting 'offline' changes. Here, we present a TUS protocol that modulates brain activation in macaques for more than one hour after 40 s of stimulation, while circumventing auditory confounds. Normally activity in brain areas reflects activity in interconnected regions but TUS caused stimulated areas to interact more selectively with the rest of the brain. In a within-subject design, we observe regionally specific TUS effects for two medial frontal brain regions – supplementary motor area and frontal polar cortex. Independently of these site-specific effects, TUS also induced signal changes in the meningeal compartment. TUS effects were temporary and not associated with microstructural changes.
DOI: https://doi.org/10.7554/eLife.40541.001

## Introduction

In neuroscience, to understand brain circuits a two-pronged approach, entailing both recording and manipulating brain activity, is essential. In recent years, there has been extensive progress in this field, which was in part made possible by the availability of new technologies (*Bestmann and Walsh, 2017*; *Dayan et al., 2013*; *Polanía et al., 2018*). While techniques for transiently manipulating

**eLife digest** Ultrasound is well known for making visible what is hidden, for example, when giving parents a glimpse of their child before birth. But researchers are now using these high-frequency sound waves – beyond the range of human hearing – for a wholly different purpose: to manipulate the activity of the brain. Conventional brain stimulation techniques use electric currents or magnetic fields to alter brain activity. These techniques, however, have limitations. They can only reach the surface of the brain and are not particularly precise. By contrast, beams of ultrasound can be focused at a millimetre scale, even deep within the brain. Ultrasound thus has the potential to provide new insights into how the brain works.

Most studies of ultrasound stimulation have looked at what happens to the brain during the stimulation itself. But could ultrasound also induce longer-lasting changes in brain activity? Changes that persist after the stimulation has ended would be valuable for research. They would also make it more likely that we could use ultrasound to treat brain disorders by changing brain activity.

Verhagen, Gallea et al. used a brain scanner to measure brain activity in macaque monkeys after ultrasound stimulation. The results showed that 40 seconds of repetitive ultrasound changed brain activity for up to two hours. Ultrasound caused the stimulated brain area to interact more selectively with the rest of the brain. Notably, only the stimulated area changed its activity in this way. This helps rule out the possibility that the changes reflect non-specific effects. If the monkeys had been able to hear the ultrasound, for example, it would have changed the activity of the parts of the brain related to hearing. Most important of all, the changes were reversible and did not harm the brain.

The results of Verhagen, Gallea et al. show that repetitive ultrasound can induce long-lasting alterations in brain activity. It can target areas deep within the brain, including those that are out of reach with other techniques. If this procedure also shows longer-lasting effects in people, it could yield valuable insights into the links between brain and behaviour. It could also help us develop new treatments for neurological and psychiatric disorders.

DOI: https://doi.org/10.7554/eLife.40541.002

activity in rodents, such as microstimulation, optogenetics, and chemogenetics (*Sternson and Roth, 2014*; *Vanduffel, 2016*), are increasingly accessible and applied, techniques for manipulating activity in the primate brain are less widely available and remain accessible to comparatively few researchers in a limited number of research centres worldwide (*Galvan et al., 2018*; *Krug et al., 2015*; *Vanduffel, 2016*). A prominent limitation for many brain stimulation tools for research and therapeutic interventions is the duration of the induced neuromodulatory effects, often not outlasting the stimulation by more than a few seconds or minutes.

Here we report on a particular protocol of low intensity pulsed transcranial focused ultrasound stimulation (TUS) that we show induces a sustained period of neuromodulation in primates without inducing structural damage. The TUS approach in general is a relatively new one (*Dallapiazza et al., 2018*; *Tufail et al., 2011*; *Tyler et al., 2018*; *Yoo et al., 2011*; *Younan et al., 2013*) and like transcranial magnetic stimulation (*Dayan et al., 2013*) and transcranial electric stimulation (*Polanía et al., 2018*) it can be applied in the absence of a craniotomy. In vitro recordings have identified several mechanisms by which ultrasound stimulation could affect neurons. It has been proposed that the sound pressure wave exerts a mechanical effect on neuronal activity through ion channel gating and changes to the membrane capacitance (*Blackmore et al., 2018*; *Kubanek et al., 2018*; *Kubanek et al., 2016*; *Prieto et al., 2013*). While the precise mechanism is being determined, early applications, including the current results, suggest TUS may be suitable as a tool for focal manipulation of activity in many brain areas in primates (*Fomenko et al., 2018*; *Lee et al., 2016a*; *Legon et al., 2018*; *Munoz et al., 2018*; *Naor et al., 2016*; *Tufail et al., 2011*; *Tyler et al., 2018*; *Yoo et al., 2011*). In the macaque, its application over the frontal eye field (FEF) affects the same aspects of oculomotor behaviour that are compromised by FEF lesion, whilst leaving intact those aspects of oculomotor behaviour that are unaffected by FEF lesion (*Deffieux et al., 2013*). Pioneering work in humans has focused on modulating and eliciting evoked responses by focused ultrasound in both cortical and subcortical sensorimotor regions (*Lee et al., 2016a*; *Lee et al., 2015*; *Legon et al., 2018*; *Legon et al., 2014*). To date ultrasonic applications are primarily focused on

direct 'online' effects. Nonetheless, some studies have observed neuromodulatory effects outlasting the sonication by several minutes before returning to baseline, with more pronounced effects for protocols delivered at higher intensities ($I_{sppa}$ >5 W/cm2) and higher duty cycles (>5%; *Kim et al., 2015*; *Yoo et al., 2011*). For example, following ultrasound protocols with a relatively high rate of acoustic energy deposition, characterized by pulses tens of milliseconds long, repeated at 10 Hz, the neuromodulatory effects outlast the sonication period: for up to 10 min following 43.7 ms long 1.14MHz pulses repeated at 10 Hz for 40 s (*Dallapiazza et al., 2018*), or for up to 20 min following 30 ms long 350 kHz pulses repeated at 10 Hz for 40 s (*Ahnine et al., 2018*). We have built on such protocols when designing the current experiment.

Recent work has highlighted the possibility that in rodents some TUS protocols can evoke a startle response when the stimulation is modulated at audible frequencies (*Guo et al., 2018*; *Sato et al., 2018*). Importantly, this auditory effect is limited to the stimulation period and dissipates within 75 ms to 4 s. This work has also emphasized that in more deeply anaesthetized animals, when the intrinsic neural activity of a system is suppressed, some TUS protocols might fail to evoke action potentials at the site of stimulation. This suggests TUS' actions might be primarily neuromodulatory in nature and/or that they are most prominently observable when they interact with ongoing physiological activity. However, as has been noted (*Airan and Butts Pauly, 2018*) the impact that any stimulation protocol might have, will be a function of the animal model being used and the precise details of the ultrasound frequencies, pulse shape, protocol, and ongoing brain activity. With the offline protocol and anaesthesia regime we used we control for such potential artefacts and show that TUS has an effect that cannot be attributed to them.

Here we focused on the effects of TUS outlasting the stimulation period, investigating the impact of 40 s trains of TUS on measurements of neural activity in three macaque monkeys provided by functional magnetic resonance imaging (fMRI) up to 2 hr after stimulation (*Figure 1*, top panel). FMRI is one of the most widely used methods for estimating neural activity. Despite limitations in its spatial and temporal resolution, fMRI remains important because it is non-invasive and can often be used to provide information about activity throughout the whole brain. Rather than providing a direct measure of neural activity, however, it provides an estimate of how activity changes in tandem with sensory, cognitive, or motor events or with activity in another brain region. Typically, fMRI-measured activity in any given brain area is a function of activity in other brain areas, especially those with which it is closely interconnected (*Neubert et al., 2015*; *O'Reilly et al., 2013*). Although many brain areas may share any given individual connection (for example both areas A and B may project to C), the overall pattern of connections of each area is unique (*Passingham et al., 2002*); as such, the overall pattern of connections therefore constitutes a 'connectional fingerprint'. As a result it is possible to use fMRI measurements of correlations in the blood oxygen level dependent (BOLD) signal across brain regions to estimate the connectivity fingerprints of a given brain area (*Figure 1*, bottom right panel; *Caspari et al., 2018*; *Ghahremani et al., 2017*; *Margulies et al., 2016*, *Margulies et al., 2009*; *Mars et al., 2013*; *Sallet et al., 2013*; *Shen, 2015*; *Shen et al., 2015*). This implies that activity in any given brain area is a function of the activity in the areas with which it is interconnected.

We exploited this feature of activity to examine the impact of TUS application to two brain regions in the frontal cortex: supplementary motor area (SMA; experiment 1) and frontal polar cortex (FPC; experiments 2 and 3). Simulations showed we were able to selectively target these regions (*Figure 2*; see *Acoustic and thermal modelling* for more details). These regions have distinct anatomical and functional connections; SMA is most strongly coupled with the sensorimotor system, while the FPC interacts primarily with the prefrontal cortex and only interacts indirectly with the sensorimotor system via SMA (*Petrides and Pandya, 2007*; *Sallet et al., 2013*). This allows us to test the spatial and connectional specificity of TUS effects. In the control state, each area's activity is normally a function of the activity in the areas that constitute its connectional fingerprint. If this pattern is altered by TUS in a manner that is dependent on the location of the stimulation, then this will constitute evidence that TUS exerts a spatially selective effect on neural activity. In a within-subject design, the same three animals participated in experiments 1 and 2, and a control experiment conducted in the absence of TUS (*Figure 1*). As such, fMRI was acquired in three conditions for all animals: following SMA TUS, following FPC TUS, and in a control state. In turn, each of the MRI sessions consisted of three consecutive runs. We also validated the results of experiment two in a new set of three different animals (experiment 3).

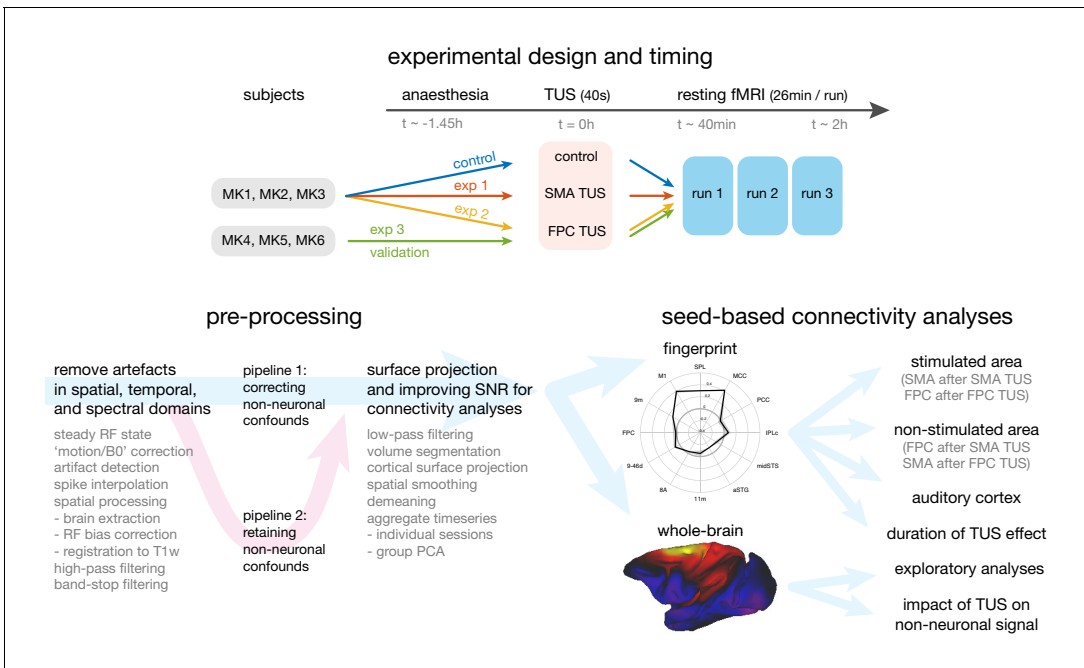

**Figure 1.** Overview of the experimental design, time-line, pre-processing pipeline, and analysis strategy. Top panel: The same three macaque monkeys (MK1, MK2, MK3) participated in experiments 1 and 2, and a control experiment conducted in the absence of TUS. Three other monkeys (MK4, MK5, MK6) participated in experiment 3. In all experiments anaesthesia was induced ~1.45 hr before the TUS intervention or no-TUS control. In experiment 1 the 40 s TUS protocol was delivered over the supplementary motor area (SMA), while in experiments 2 and 3, the protocol was targeted at frontal polar cortex (FPC). In all sessions, three consecutive runs of resting fMRI (26 min per run) were acquired starting ~40 min after the TUS or control protocol. Bottom left panel: Data from the resting fMRI runs were pre-processed following a standardized pipeline to address artefacts, improve image quality and signal-to-noise ratio, and prepare for connectivity analyses. By default, non-neuronal confounds were removed from the timeseries, but to allow an analysis of the effect of TUS on non-neuronal signal the data was also processed in parallel in a second pipeline, distinct from the default by omitting the non-neuronal confound regression procedure. Bottom right panel: The effect of TUS on the coupling patterns of the stimulated regions were quantified as 'connectional fingerprints' employing pre-defined targets (here illustrated for the SMA in the control state). These fingerprint analyses allowed us to test for the presence and duration of TUS effects in the stimulated areas (SMA after SMA TUS; FPC after FPC TUS), and in control areas, including non-stimulated areas (FPC after SMA TUS; SMA after FPC TUS) and the auditory cortex. Whole brain connectivity maps supported exploratory analyses and an assessment of the impact of TUS on non-neuronal signal (here illustrated for SMA in the control state).
DOI: https://doi.org/10.7554/eLife.40541.003

Ultrasound modulation of SMA and FPC led to focal, area-specific changes in each stimulated region's connectivity profile. In each area, a region's activity pattern after TUS application was more a function of its own activity and that of strongly connected regions, but less a function of activity in more remote and weakly interconnected areas. Independent of these specific grey matter signal changes, TUS also interacted with non-neuronal structures, as evidenced by more widespread signal changes observed in the meningeal compartment (including cerebral spinal fluid and vasculature). Finally, in experiment four we demonstrate that TUS application had no observable impact on cortical microstructure apparent on histological examination. In summary, TUS over dorsomedial frontal cortex causes spatially specific sharpening of the stimulated region's connectivity profile with high efficacy and reproducibility, independent of non-neuronal signal changes.

## Results

### Experiment 1, TUS modulation of SMA connectivity

Transcranial focused ultrasound stimulation of SMA induced spatially specific changes to the connectivity profile of SMA. At rest, in the control state, SMA's activity is coupled with activity throughout sensorimotor regions in frontal and parietal cortex, inferior parietal, prefrontal, and parts of cingulate cortex (*Figure 3a*). Many of these regions are anatomically connected with SMA (*Dum and*

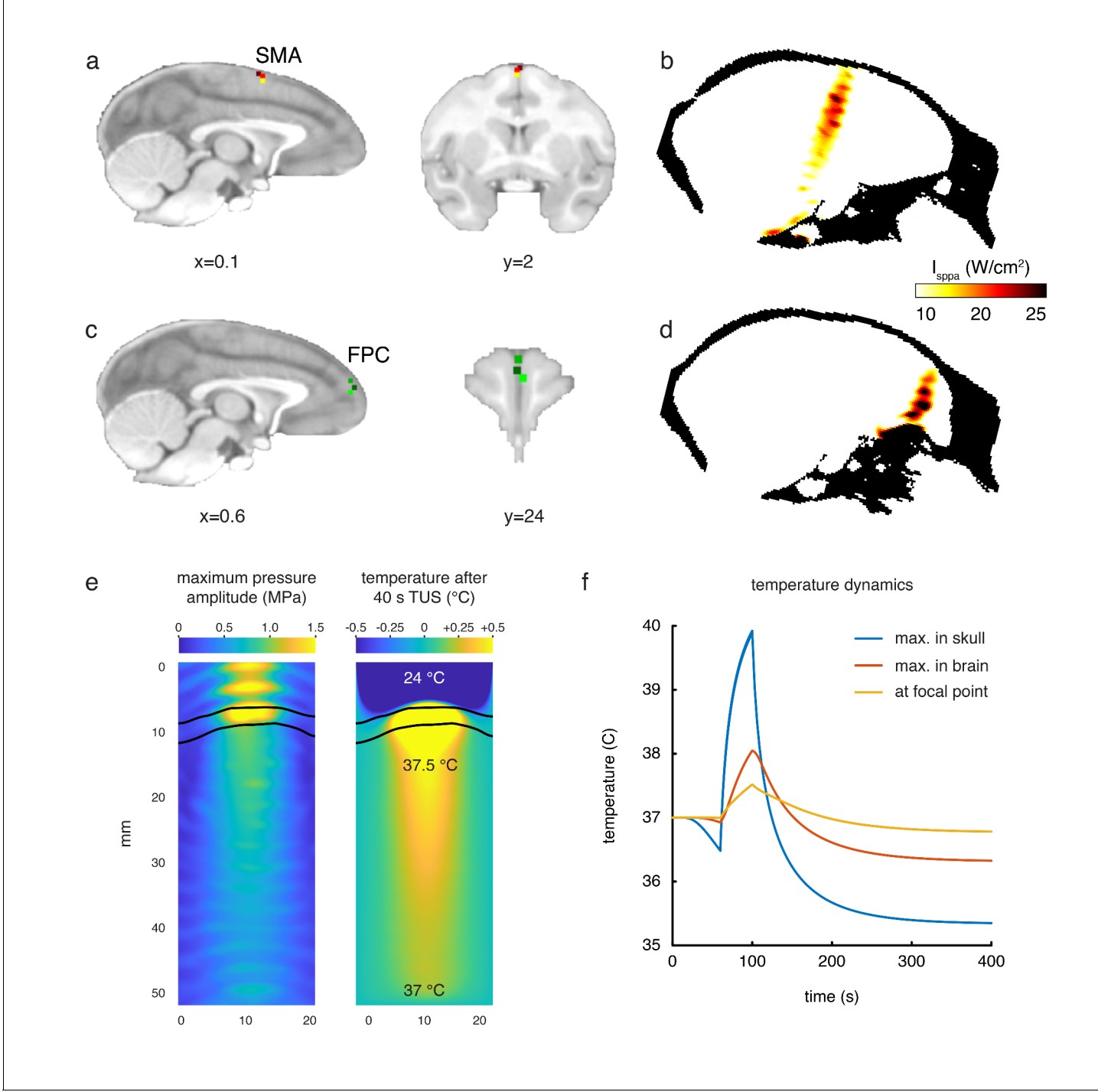

**Figure 2.** Stimulation targets and thermal modelling. (**a**) Stimulation target positions in SMA are shown for each of the three individual animals in three different colours on sagittal and coronal views. (**b**) Estimates of the focused ultrasound peak intensities and spatial distribution when targeting SMA are derived from numerical simulations using a high-resolution macaque whole-head CT scan, here displayed on a midline sagittal section. (**c**) FPC targets in three animals are shown on sagittal and coronal sections. (**d**) Estimated peak intensities and spatial distribution when targeting FPC shown on a sagittal section. (**e**) Whole-head simulations of the acoustic wave and thermal dynamics provided estimates of the maximum pressure amplitude (left panel) and the temperature after 40 s TUS (right panel). The data depict a cropped plane of the whole-head simulations with the sonic coupling cone at the top and the brain at the bottom; the skull is outlined in black. Pressure and temperature are maximal in the skull, which is more absorbing than soft tissue. (**f**) Temperature dynamics for the maximum temperature in the skull (blue), maximum temperature in the brain (red) and at the geometrical focal point in the cortex (yellow).

DOI: https://doi.org/10.7554/eLife.40541.004

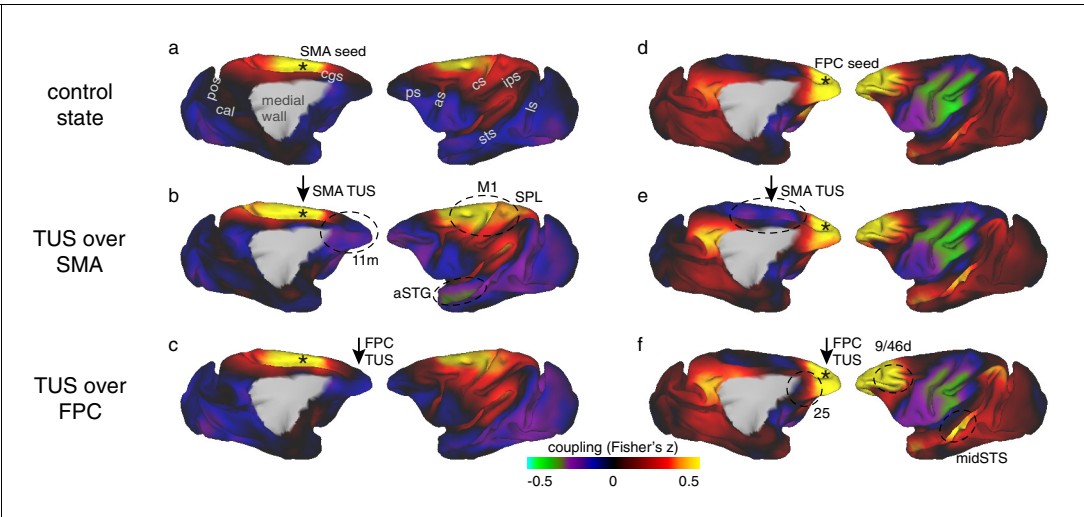

**Figure 3.** Coupling of activity between stimulated areas and the rest of the brain in experiments 1 (SMA) and 2 (FPC). The left panels show activity coupling between SMA and the rest of the brain in the control state (a), after SMA TUS (b), and after FPC TUS (c). The right panels show activity coupling between FPC and the rest of the brain in the control state (d), after SMA TUS (e), and after FPC TUS (f). Functional connectivity from TUS-targeted regions is therefore summarized in panels (b) and (f) (i.e. SMA connectivity after SMA TUS and FPC connectivity after FPC TUS). Each type of TUS had a relatively selective effect on the stimulated area: SMA coupling was changed by SMA TUS (b) and FPC coupling was changed by FPC TUS (f). Positive correlations are represented in warm colours from red to yellow, negative correlations are represented in cool colours from blue to green. Key regions of change are highlighted by black dashed ovals. TUS target sites are indicated with arrows. Connectivity seed regions are indicated with black asterisks. Key anatomical features are labelled in panel (a): pos, parieto-occipital sulcus; cal, calcarine sulcus; cgs, cingulate sulcus; ps, principal sulcus; as, arcuate sulcus; cs, central sulcus; ips, intraparietal sulcus; sts, superior temporal sulcus; ls, lunate sulcus.
DOI: https://doi.org/10.7554/eLife.40541.005

The following figure supplement is available for figure 3:

**Figure supplement 1.** Specific patterns of change in the coupling of activity between stimulated areas and the rest of the brain were replicated in experiments 2 and 3.
DOI: https://doi.org/10.7554/eLife.40541.006

*Strick, 2005*; *Geyer et al., 2000*; *Strick et al., 1998*). If we change the responsiveness of SMA neurons to such activity in interconnected regions by artificially modulating SMA activity with TUS, then we should see a change in the coupling between SMA activity and activity in other regions. Following ultrasound stimulation, SMA changed its coupling with the sensorimotor system, anterior and posterior cingulate, anterior temporal, inferior parietal, and prefrontal cortex (*Figure 3b*). This can be seen on the whole brain functional connectivity maps for the SMA region (*Figure 3*, compare panels a and b, representative changes highlighted by dashed black circles) and on the whole brain differential connectivity maps in *Figure 4* (panels b and c).

It is also apparent in the illustration of SMA's connectional fingerprint (*Figure 5a*). The distance of each coloured line from the centre of the figure (and hence its proximity to the circumference of the figure) indicates the strength of activity coupling between SMA and each of the other brain areas indicated on the circumference. Compared to the control state (blue line), after TUS over SMA (red line), SMA's positive coupling is enhanced with proximal areas in the sensorimotor system but reduced in many long-range connections (non-parametric permutation test, p=0.017). The primary motor cortex (M1), superior parietal lobe (SPL), and middle cingulate cortex (MCC) in the dorsomedial sensorimotor network have been reported to be closely connected with the SMA, whereas prefrontal regions on the dorsomedial (area 9m and FPC), dorsolateral (areas 9-46d and 8A), and ventromedial (area 11m) surface, and those in the temporal lobe (anterior superior temporal gyrus, aSTG; middle superior temporal sulcus, midSTS), and parietal cortex (caudal inferior parietal lobule, IPLc; posterior parietal cortex, PCC) have been reported to be less closely connected with the SMA (*Dum and Strick, 2005*; *Geyer et al., 2000*; *Strick et al., 1998*). TUS increased positive coupling between the stimulated area and proximal areas normally closely connected with it, while, at the same time, decreasing coupling between the stimulated area and many areas normally less closely

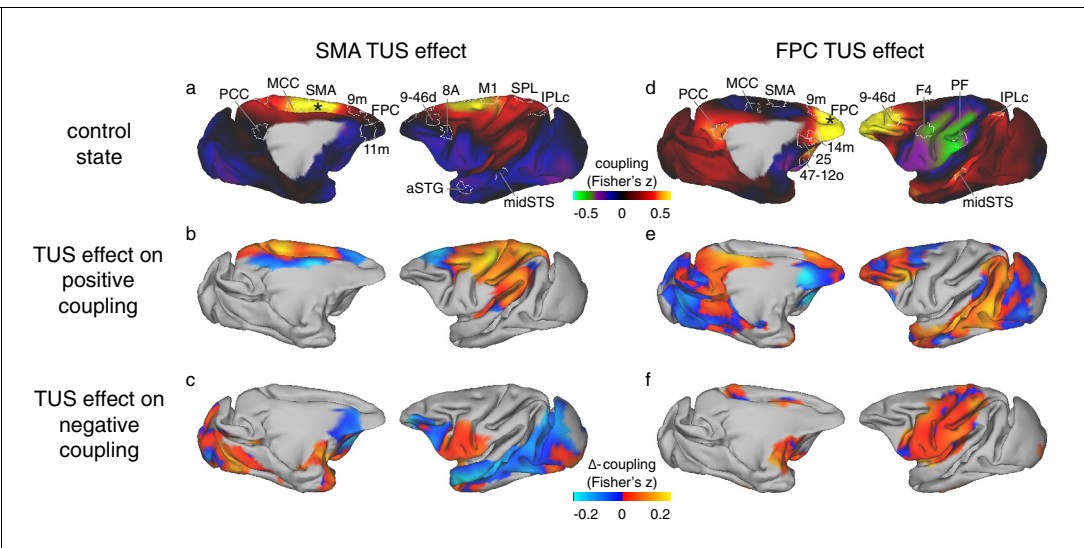

**Figure 4.** Differential effect of TUS on coupling of activity between stimulated areas and the rest of the brain in experiments 1 (SMA) and 2 (FPC). The left panels show activity coupling between SMA and the rest of the brain in the control state (a), the differential effect of SMA TUS for areas positively coupled (z > 0.1) with SMA in the control state (b), and for areas negatively coupled (z < −0.1) with SMA in the control state (c). The right panels show activity coupling between FPC and the rest of the brain in the control state (d), the differential effect of FPC TUS for areas positively coupled (z > 0.1) with FPC in the control state (e), and for areas negatively coupled (z < −0.1) with FPC in the control state (f). Panels (a) and (b) are reproduced here from *Figure 3* for reference and to illustrate the location and extent of the ROIs used in the fingerprint analyses (*Figure 5*). Hot colours in (b) and (e) indicate enhanced coupling following TUS compared to the control state, while cool colours indicate reduced coupling. In (c) and (f) hot colours indicate reduced negative coupling, while cool colours indicate further negative coupling. All other conventions as in *Figure 3*. The TUS induced changes to the coupling of the stimulated regions were not limited to the *a-priori* defined ROIs but extended across many regions according to the connectional topography of the stimulated region.
DOI: https://doi.org/10.7554/eLife.40541.007

connected with it. This pattern not only emerges from the fingerprint analyses, but constitutes a principle evident across the brain, as illustrated by whole-brain differential SMA-connectivity maps of the effect of SMA TUS (*Figure 4*).

These specific effects of TUS were sustained over the duration of our experiment, lasting up to 2 hr (*Figure 5d–f*). Disruptive effects of TUS on long-range coupling were especially prominent immediately following the end of TUS application (*Figure 5d*), and gradually reduced towards the end of our recording session (*Figure 5f*). The enhancing effects that TUS exerted on SMA's coupling with adjacent and strongly connected areas had a relatively delayed appearance, arising well after the TUS had ended (more than 1 hr, *Figure 5e*), but again decreasing towards the end of the recording session. In summary, the most important finding was of a protracted period of connectivity change after TUS. While we are cautious about overinterpreting precise timing differences between long-range connectivity reductions and local connectivity increments, we note that the observed pattern could signify distinct time courses for TUS-induced long-term depression and long-term potentiation. However, the observed pattern is also consistent with the notion that early disruption of long-range input to a network (*Figure 5d*) leaves relatively more signal variance in this network to be explained by remaining local input, driving the observation of subsequent within-network coupling increments (*Figure 5e*). Both these mechanisms would lead to a sharpening of the stimulated region's connectivity profile, as observed here.

## Experiment 2, TUS modulation of FPC connectivity

Like SMA, FPC's activity is coupled with that in interconnected brain regions even when animals are at rest in the control state (*Figure 3d*). FPC's activity is positively correlated with activity in a number of adjacent dorsomedial and lateral prefrontal areas and in the central portion of the superior temporal sulcus (midSTS) and posterior cingulate cortex with which it is monosynaptically interconnected

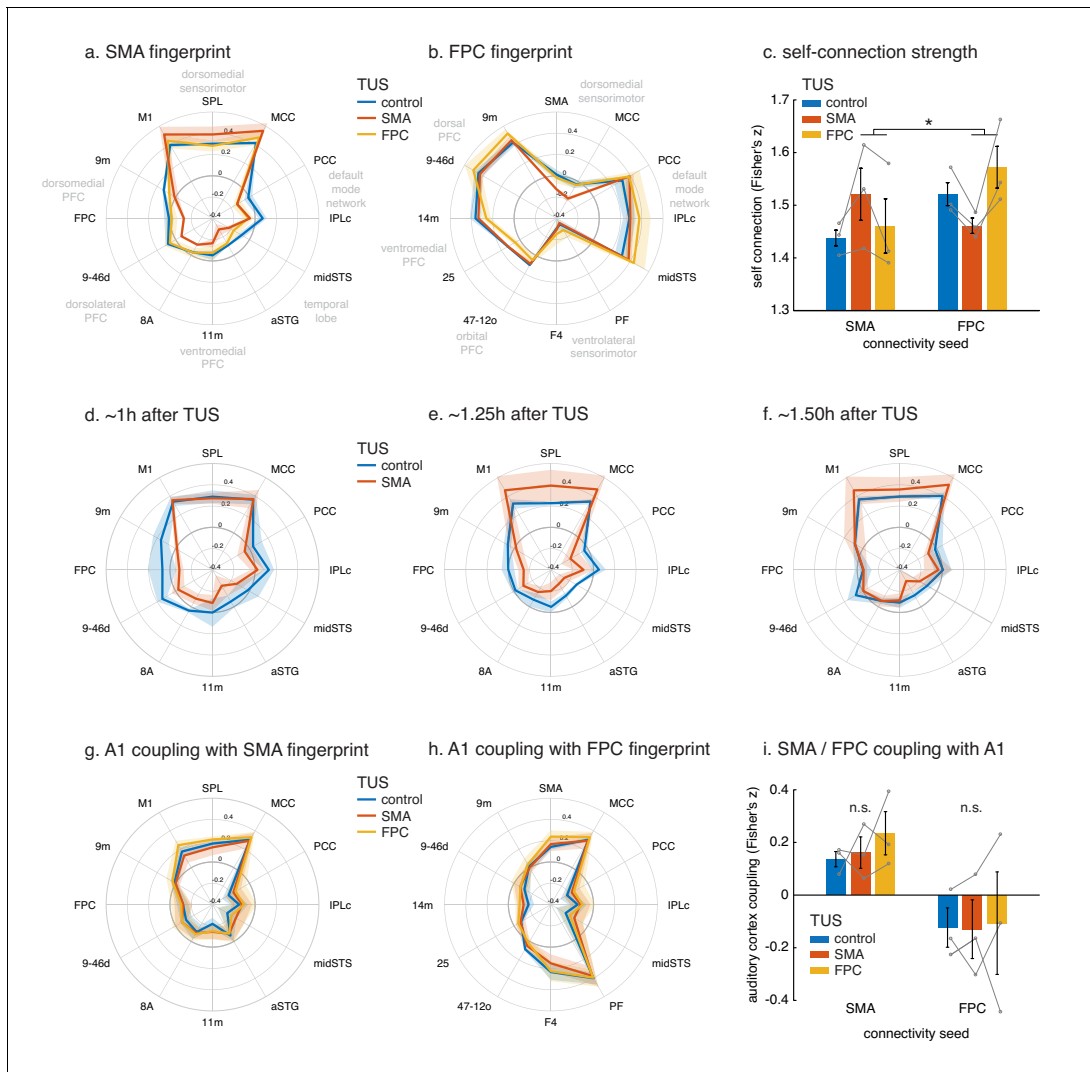

**Figure 5.** Connectional fingerprints of the stimulated areas. Fingerprint target regions were drawn from the literature and chosen based on their known and distinct connectivity, either strong or weak, with the seed region. Activity coupling in the control state is indicated in blue, following SMA TUS in red, and FPC TUS in yellow. (**a**) SMA connectional fingerprint. In the control state (blue) SMA is strongly coupled to areas in the dorsomedial sensorimotor system (M1, SPL, MCC). Coupling with each of these areas is enhanced after SMA TUS (red). In contrast, coupling with regions that SMA is weakly connected with in the control state are even further reduced after SMA TUS (red). However, SMA's fingerprint is relatively unaffected after FPC TUS (yellow), but some effects are visible for regions that are strongly coupled with FPC (compare to panel b). (**b**) FPC connectional fingerprint is sharpened following FPC TUS, but not following SMA TUS. The latter only affected coupling with SMA itself and regions strongly connected to it (MCC). (**c**) TUS induced more homogenous activity within the stimulated area (SMA in red on the left, FPC in yellow on the right). (**d–f**) Temporal evolution of the effect of SMA TUS on the SMA connectional fingerprint. The left panel depicts effects observed in the first fMRI run, followed by the second run in the middle, and the third and last run on the right. For each run, the time after TUS refers to the duration between the end of TUS and the midpoint of the run, averaged across the three animals. (**g–h**) TUS-induced effects on the connectional fingerprints of SMA and FPC could not be explained by auditory activity. (**i**) There were no changes in the coupling between the stimulation sites and auditory cortex. In all plots, thick lines and histograms indicate mean activity coupling; lighter coloured error bands and error bars depict standard-error of the mean; the asterisk denotes the interaction between TUS and connectivity seed (p<0.001).

DOI: https://doi.org/10.7554/eLife.40541.008

The following figure supplement is available for figure 5:

**Figure supplement 1.** TUS did not affect the temporal variability of the BOLD signal.

DOI: https://doi.org/10.7554/eLife.40541.009

(*Petrides and Pandya, 2007*). By contrast there was, at a rest, a negative relationship with the activity in sensorimotor areas, with which FPC is indirectly connected via regions such as SMA.

In comparison to the control state, FPC stimulation induced an enhancement in the normal short-range connectivity between FPC and adjacent dorsomedial (area 9m) and lateral prefrontal cortex (area 9-46d) with which it is particularly strongly connected. In addition, a similar effect was seen in more distant regions with which it is also strongly connected – the midSTS, IPLc and PCC, together comprising temporal and parietal segments of the primate 'default mode network' (*Petrides and Pandya, 2007*). By contrast, there was reduced coupling with other areas in prefrontal cortex, including ventromedial (area 14m), subgenual cingulate (area 25), and lateral orbitofrontal cortex (area 47-12o). These are all areas that FPC is connected to but less strongly (*Petrides and Pandya, 2007*). Finally, TUS applied to FPC also led to a change in long-range connectivity between FPC and several motor association regions with which it is not directly connected, especially those in the ventrolateral parieto-frontal sensorimotor network (areas PF and F4 *Petrides and Pandya, 2007*). As noted, in the control state, the activity in FPC and these sensorimotor association areas is negatively or anti-correlated, but this anti-correlation was reduced by FPC TUS. These results are apparent in the whole brain functional connectivity maps for the FPC region (*Figure 3*, compare panels d and f, representative changes highlighted by dashed black circles) and on the whole brain differential connectivity maps in *Figure 4* (panels d and f). It is also apparent in the illustration of FPC's connectional fingerprint (*Figure 5b*). Here the blue line indicates the strength of activity coupling between FPC and each of the other brain areas indicated on the circumference in the control state. The yellow line shows that FPC's coupling with each area is changed after FPC TUS (non-parametric permutation test, p=0.027).

## Comparing experiments 1 (SMA) and 2 (FPC)

It is important to test the claim that TUS induces effects that are spatially specific to each sonicated area by directly comparing effects between stimulation sites. Although FPC TUS significantly altered FPC functional connectivity, it had comparatively little impact on SMA's pattern of functional connectivity; there was no difference in SMA's functional connectivity between the control state and after FPC TUS (non-parametric permutation test, p=0.231; whole-brain map in *Figure 3c* and yellow line in connectivity fingerprint in *Figure 5a*). Importantly, the effects of TUS over SMA on SMA's connectivity were significantly dissociable from the effects of FPC TUS (non-parametric permutation test, p=0.041). Similarly, SMA TUS had some but comparatively little impact on FPC's pattern of functional connectivity (non-parametric permutation tests, SMA versus control, p=0.047; SMA versus FPC, p=0.028; whole-brain map in *Figure 3e* and red line in connectivity fingerprint in *Figure 5b*). In fact, the most prominent changes in each area's connectional fingerprint that were induced by stimulation of the other area were the disruption of functional connectivity between FPC and sensorimotor areas when SMA was stimulated (*Figure 3e*, encircled). This particular result may have occurred because, as already noted, FPC has no direct monosynaptic connections with these sensorimotor areas (*Petrides and Pandya, 2007*) and so its functional coupling with these areas is likely to be mediated by areas such as SMA and the areas that surround it such as the pre-supplementary motor area and the cingulate motor areas (*Bates and Goldman-Rakic, 1993*; *Lu et al., 1994*). In fact, these circumscribed exceptions may well confirm the rule that a region's connectivity pattern is only affected by stimulation of the region itself. Namely, SMA TUS only affects FPC coupling with SMA itself and the regions which are coupled with FPC only through SMA. In conclusion, the effects of TUS in different frontal regions were clearly dissociable.

## Experiment 3, replication of TUS effects on FPC connectivity

We investigated the reproducibility of TUS effects by examining the impact of TUS to FPC in three additional individuals in a biological replication experiment (experiment 3, *Figure 3—figure supplement 1*). TUS had the same effects as seen in experiment 2: a site-specific sharpening of the stimulated region's connectional profile. While experiments 1 and 2 followed a within-subject design, for experiment three we conducted a between-subject analysis where the subjects in the two experiments differed in age. We are therefore careful not to draw too strong conclusions on any main effect of subject group but focus on the interaction of the TUS effect with the fingerprint shape. Notwithstanding, when reviewing the simple effects driving the well-matched interaction, we note that

the observed sharpening was less prominently related to activity coupling decreases in experiment 3 than in experiment 2. Putatively, parameters of the general anaesthesia could impact on the effect of TUS. However, subjects used in experiment 2 and 3 did not differ regarding depth of anaesthesia or duration between sedation and fMRI data collection.

## Effects of TUS on activation in the stimulated regions and on the auditory system

It has recently been suggested that certain TUS protocols might have a limited efficacy in evoking spiking activity at the stimulation site, but rather exert their influence on the brain through the auditory system, not unlike an auditory startle response (*Guo et al., 2018*; *Sato et al., 2018*). Although these online observations in rodents should perhaps not be extrapolated far outside the tested conditions (for example to our measurements taken tens of minutes after the stimulation ended), these observations do argue in favour of performing controlled experiments that address and exclude such confounds (*Airan and Butts Pauly, 2018*). Here we consider, first, why neural effects of TUS might be evident in the current study when they were not clear previously. Second, we consider whether neural effects may be due to an auditory artefact.

First, it is possible that the efficacy of TUS is a function of both the specifics of the stimulation protocol and of ongoing neural activity. A neuromodulatory technique may fail to elicit spiking activity in deeply anaesthetized rodents (*Guo et al., 2018*; *Sato et al., 2018*). However, in this study we specifically test whether it simply modulates ongoing activity, while adopting lighter anaesthesia levels. Importantly, it is known that whole-brain functional connectivity, as measured with the BOLD signal, is preserved at these levels (*Mars et al., 2013*; *Neubert et al., 2015*; *Neubert et al., 2014*; *O'Reilly et al., 2013*; *Sallet et al., 2013*; *Vincent et al., 2007*). Finally, the experiments were conducted in a primate model as opposed to a rodent model; the importance of species-specific effects in TUS models are currently unknown. Under these distinct conditions, we observed that TUS modulated the activity coupling of each stimulated area in a regionally specific manner.

Nevertheless, following these investigations of the effect of TUS on whole-brain connectivity patterns of the stimulated regions, we carried out a second line of investigation and examined the effect of TUS on the signal in the stimulated regions themselves (*Figure 5c*). While BOLD fMRI cannot provide an absolute measure of neural activity, we can characterize how homogeneous the activation signal is within the stimulated region, as quantified by the coupling strength of the signal at each point in the stimulated region of interest to all other points in that region. This analysis revealed that TUS induced more spatially homogenous activation within the stimulated area, but not in the non-stimulated region (interaction of TUS x connectivity seed: $F_{(1,8)}$=1571.2, p=1.8044e-10, $d$ = 11.4426, CI=[1.2733 1.1333]). This effect on spatial homogeneity of the signal was not accompanied by changes to the temporal variance of the BOLD signal fluctuations in the stimulated or other regions (*Figure 5—figure supplement 1*). In fact, the standard deviation of the BOLD signal fluctuations over time were strikingly similar between the two stimulation sites (SMA and FPC, highlighted in *Figure 5—figure supplement 1*) and across the different experimental conditions (control, SMA TUS, and FPC TUS). This suggests that TUS leaves intact basic haemodynamics and neurophysiology and instead has a circumscribed and specific impact on the coupling of the stimulated region with the rest of the brain.

Third, the presence of auditory and somatosensory confounds is likely to be a function of the specifics of the TUS protocol. In sonication protocols the ultrasound wave is often pulse modulated at ~1 kHz, well within the audible range of many rodents and primates. At these modulation frequencies auditory stimulation is perhaps not unexpected (*Guo et al., 2018*; *Sato et al., 2018*). This is something that we have avoided in our work with macaques: we pulse modulated the 250 kHz ultrasound wave at 10 Hz: as such we ensured that the frequency of both the ultrasound wave and its modulating envelope are well outside of the macaque hearing range. Moreover, here we adopted an offline experimental design where any potential audible stimulation associated with the TUS application was limited to the 40 s sonication period, while the neural activation measures were initiated tens of minutes later. Furthermore, the specificity of our results strengthens the suggestion that it might not be possible to explain away the current findings as the result of an auditory artefact having occurred up to two hours earlier.

Nevertheless, we also carried out a fourth line of inquiry and examined the activation in the primary auditory cortex (A1) and its relationship with activity in the rest of the brain (*Figure 5g–i*). The

effects of TUS on SMA and FPC coupling patterns, as quantified in their connectional fingerprints, could not be explained by a potential impact of TUS on A1 activity coupling. In fact, TUS did not affect A1 activity coupling with SMA fingerprint targets (*Figure 5g*; non-parametric permutation tests, SMA TUS: p=0.8234, FPC TUS p=0.3452), nor its coupling with FPC fingerprint targets (*Figure 5h*; non-parametric permutation tests, SMA TUS: p=0.5411, FPC TUS p=0.2667). Moreover, neither SMA nor FPC changed its coupling with A1 as a function of TUS (*Figure 5i*; main effect of TUS: $F_{(1,8)}$=0.015445, p=0.90416, $d$ = 0.03583, CI=[−0.41483 0.46209]; interaction of TUS x connectivity seed: $F_{(1,8)}$=0.18284, p=0.68022, $d$ = 0.1234, CI=[−0.25058 0.36466]).

## Non-neuronal signal changes

The ability of ultrasound to reversibly interact with biological tissue is not limited to grey matter. We were aware that our ultrasonic beam, placed over the central midline to target SMA or FPC in both hemispheres simultaneously, was also likely to reach the meningeal compartment in the interhemispheric fissure. In fMRI analyses this region is sometimes referred to as 'cerebral spinal fluid', although in reality it contains the cortical membranes (dura, arachnoid, and pia mater), some cerebral spinal fluid, and important vascular structures, such as the superior sagittal sinus. Ultrasound protocols designed to induce vasodilation or to temporally open the blood-brain-barrier are conventionally markedly distinct from those employed here, for example using higher intensities or supplemented with intravenously injected microbubbles. Nonetheless, we set out to test the influence of ultrasound on what we shall continue to refer to as the 'meningeal' signal and the grey matter signal. To do this it was obviously necessary to take a somewhat unconventional rs-fMRI analysis approach that did not remove the meningeal signal that is typically regarded as a confound.

A principal component analysis of the signal in the meningeal and grey matter compartments revealed the main components in either compartment explained significantly more variance following TUS compared to control (*Figure 6c*; main effect of TUS: $F_{(1,9)}$=30.6, p=0.00036, $d$ = 0.67031, CI=[2.1916 5.3343]; in grey matter: $F_{(1,4)}$=10.743, p=0.0306, $d$ = 1.3381, CI=[0.47735 5.7655]; in meningeal compartment: $F_{(1,4)}$=16.263, p=0.0157, $d$ = 1.6464, CI=[1.3379 7.2512]). The fact that this effect is present in both compartments could reflect the tight vascular coupling between grey matter and meningeal signal or be driven by partial-voluming effects (these are more pronounced when the size of the brain is relatively small, as for monkey fMRI). This observation suggests that after TUS the BOLD signal became more homogenous. In a seed-based connectivity analysis, as performed here, this would be reflected in a stronger contribution of global signal coupling. However, the impact of TUS presented above does not seem to exhibit this effect, as illustrated by the specificity of the TUS effects for SMA and FPC (*Figures 3,5*), and further underpinned by the absence of TUS effects in regions remote from the stimulation sites (for example, *Figure 6a,d* illustrates the case of the posterior parietal operculum, POp). Importantly, these coupling estimates are obtained when following the conventional rs-fMRI analysis approach to account for global signal confounds by removing WM and meningeal signal contributions before estimating grey matter coupling indices.

We hypothesized that if TUS leads to more homogenous global signal, its contribution to the grey matter signal might have been accounted for when removing meningeal signal components in a linear regression framework (*Verhagen, 2012*). Accordingly, we have repeated the seed-based connectivity analyses after accounting for global signal confounds based on the white matter compartment alone, excluding the meningeal compartment. In these data, global signal contributions were indeed preserved as evidenced by anatomically implausible global connectivity patterns present in the control state (compare panels a and b in *Figure 6*, e.g. the prefrontal cortex, encircled). As such, this procedure allowed us to interrogate the global effects of TUS over dorsomedial frontal regions on BOLD signal (see Materials and Methods for full details). Following this procedure, we observed that SMA stimulation appeared to induce widespread increases in signal coupling compared to the control state. This effect was not limited to the stimulation site but also present in remote regions (e.g. when seeded in POp, compare panels b and e in *Figure 6*). This effect can be quantified by considering the strength of local connections for every point in the cortex. It is then apparent that the changes induced by stimulation are global in nature (*Figure 6f*). In general, when not fully accounting for global confounds, a region's connectivity profile after TUS could be predicted by considering its profile in the control state and adding a spatially flat constant. This suggests an additive non-neuronal source, captured by signal components in the meningeal compartment, may explain the presence and enhancement of global signal observed following TUS

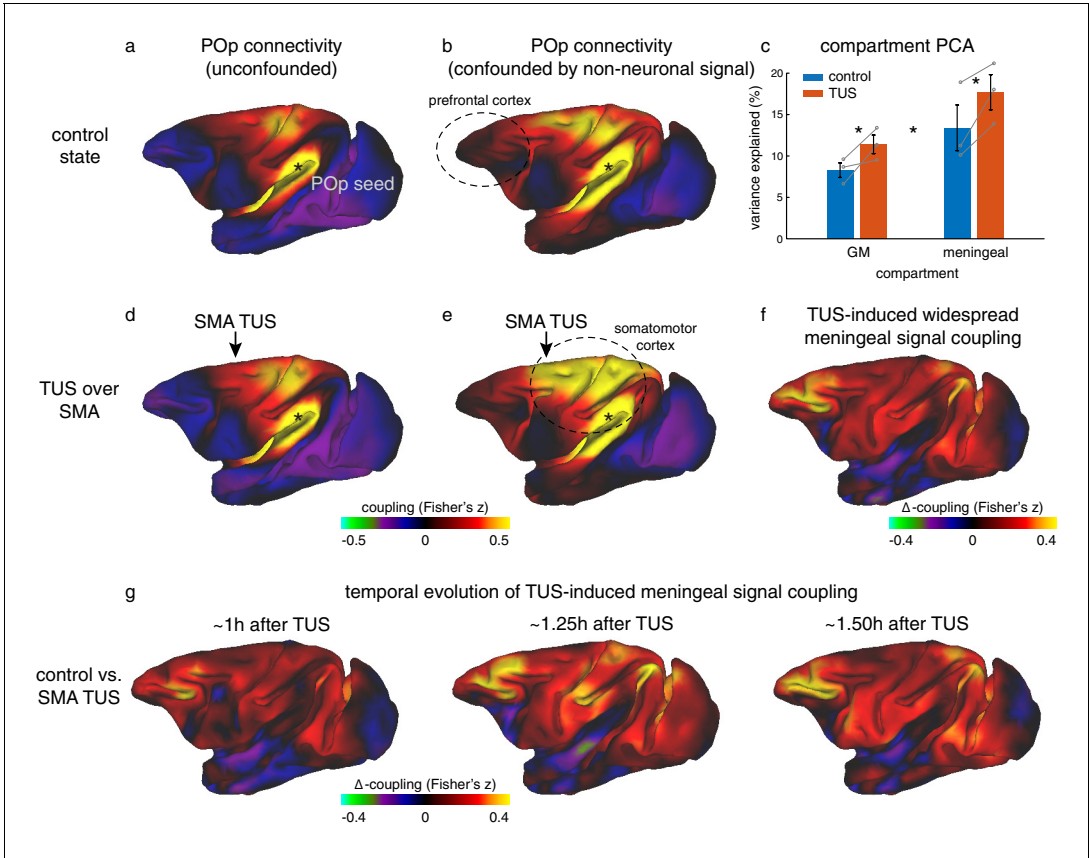

**Figure 6.** Non-neuronal signal changes driving widespread coupling. In fully processed fMRI data, unconfounded by global nuisance sources, effects of TUS were not immediately apparent outside the stimulated areas. (**a,d**) For example, the whole-brain connectivity pattern of the parietal operculum area (POp, spatially removed from either stimulation site) was unaffected by TUS over SMA. (**b**) When nuisance contributions from the meningeal compartment are not removed (but those from WM are), the presence of globally shared signal is evident: POp activity is now coupled with that in many other areas. (**e**) POp's global nuisance confounded functional connectivity pattern became stronger after SMA TUS. This was because SMA TUS induced broad changes in BOLD signal even in the meningeal compartment, captured in a principal component analysis (**c**). Error bars denote standard-error of the mean; asterisks denote effects of TUS and WM/meningeal compartment (p<0.05). (**f**) This led to many points in the cortex exhibiting stronger coupling with other brain areas; all areas shown in red are points that have stronger coupling with the rest of the brain after TUS. Note that this effect is present far beyond the stimulation site. (**g**) This widespread TUS-induced meningeal signal coupling persisted over time. All conventions as in *Figures 3,5*.

DOI: https://doi.org/10.7554/eLife.40541.010

The following figure supplement is available for figure 6:

**Figure supplement 1.** Widespread non-neuronal signal changes were replicated in experiments 2 and 3.

DOI: https://doi.org/10.7554/eLife.40541.011

over SMA (*Figure 6e,f*, e.g. primary sensorimotor cortex). These effects of TUS on widespread coupling mediated by meningeal signal persisted over time for more than 1 hr after stimulation had ended (*Figure 6g*).

Similar effects of TUS on meningeal signal were also observed after FPC TUS (*Figure 6—figure supplement 1*). We note that in unconfounded fMRI data FPC TUS did not have a strong impact outside the stimulated region (illustrated for POp in *Figure 6—figure supplement 1*). In contrast, in fMRI data confounded by meningeal-driven global nuisance, FPC stimulation led to widespread enhanced signal coupling compared to the control state. This effect of FPC TUS was replicated in a new set of animals in experiment 3 (*Figure 6—figure supplement 1*). Although the effect was perhaps not as strong as that seen after SMA TUS, this finding again suggests that TUS over the medial meningeal compartment may produce widespread changes that are non-neuronal in origin. Differences in morphology of the sagittal sinus along the rostro-caudal axis might explain this weakened global effect.

## Experiment 4, meso- and micro-structural analyses

Some higher intensity ultrasound stimulation protocols, distinct from those used here, have been shown to induce thermal lesions or haemorrhage following cavitation (*Elias et al., 2013*). Despite the fact that 40 s trains of TUS induced sustained changes in the post-stimulation period in experiments 1 and 2, no structural changes remotely resembling those seen with higher intensity ultrasound protocols were observed. First of all, we did not observe any indication of TUS-induced oedema when comparing T1w MRI structural scans collected in baseline sessions with T1w scans collected after TUS (*Figure 7*). We also did not observe tissue alteration (e.g. tissue burn) at the post-mortem examination. Neither were any signs of neuronal alteration or haemorrhage observed in histological analyses of three macaques following pre-SMA TUS (*Figure 8*).

## Acoustic and thermal modelling

To quantify the pressure amplitude, peak intensities, spatial distribution, and potential temperature changes in the monkey brain associated with the TUS protocol used in this study we simulated the acoustic wave propagation and its thermal effect in a whole head finite element model based on a high-resolution monkey CT scan. As estimated by these numerical simulations, the maximum spatial-peak pulse-averaged intensity ($I_{sppa}$) at the acoustic focus point was 24.1 W/cm$^2$ for the SMA target and 31.7 W/cm$^2$ for the FPC target (spatial peak temporal average intensities, $I_{spta}$: 7.2 W/cm$^2$ and 9.5 W/cm$^2$ for SMA and FPC, respectively). Given that the skull is more acoustically absorbing than soft tissue, the highest thermal increase is located in the skull itself, estimated by the simulation to be 2.9°C. Given an approximate 0.5 mm thickness of the dura (*Galashan et al., 2011*) the maximum temperature below the dura was 38.0°C. The maximal thermal increase at the geometrical focus of the sonic transducer was less than 0.5°C (*Figure 2*).

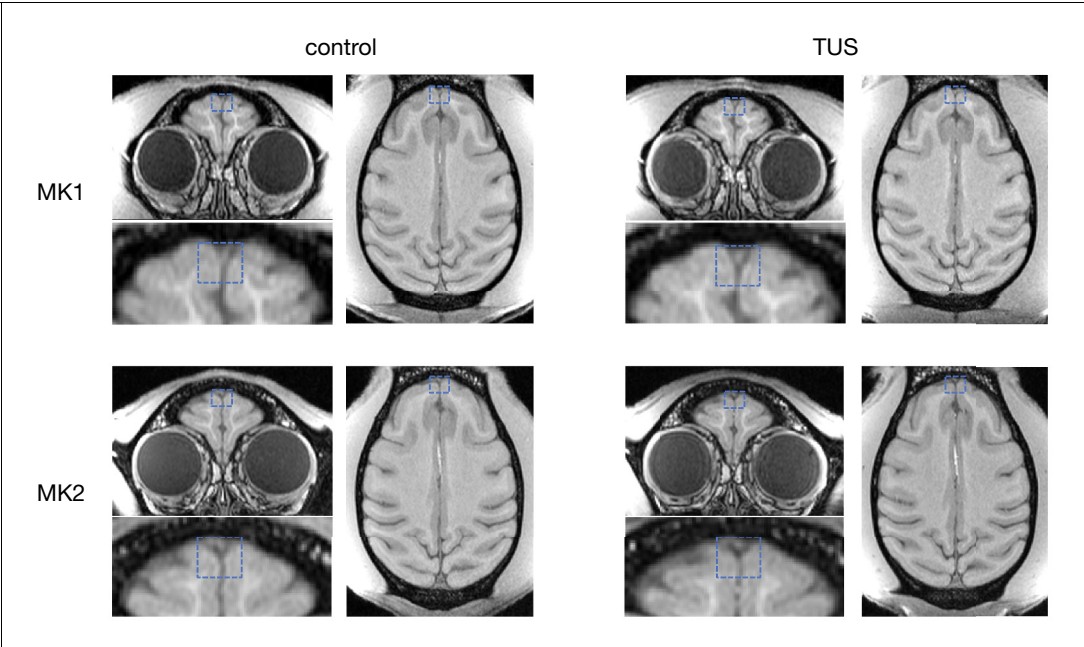

**Figure 7.** No effect of TUS was apparent on T1w structural MRI images. Examination of T1-weighted structural MRI images did not reveal any evidence for oedema after TUS. For two monkeys (top and bottom row, respectively), T1w images were acquired immediately following the resting-state fMRI runs in the control state (left column) and ~2 hr after TUS over FPC (right column). The sonication target regions are highlighted with dashed blue boxes.

DOI: https://doi.org/10.7554/eLife.40541.012

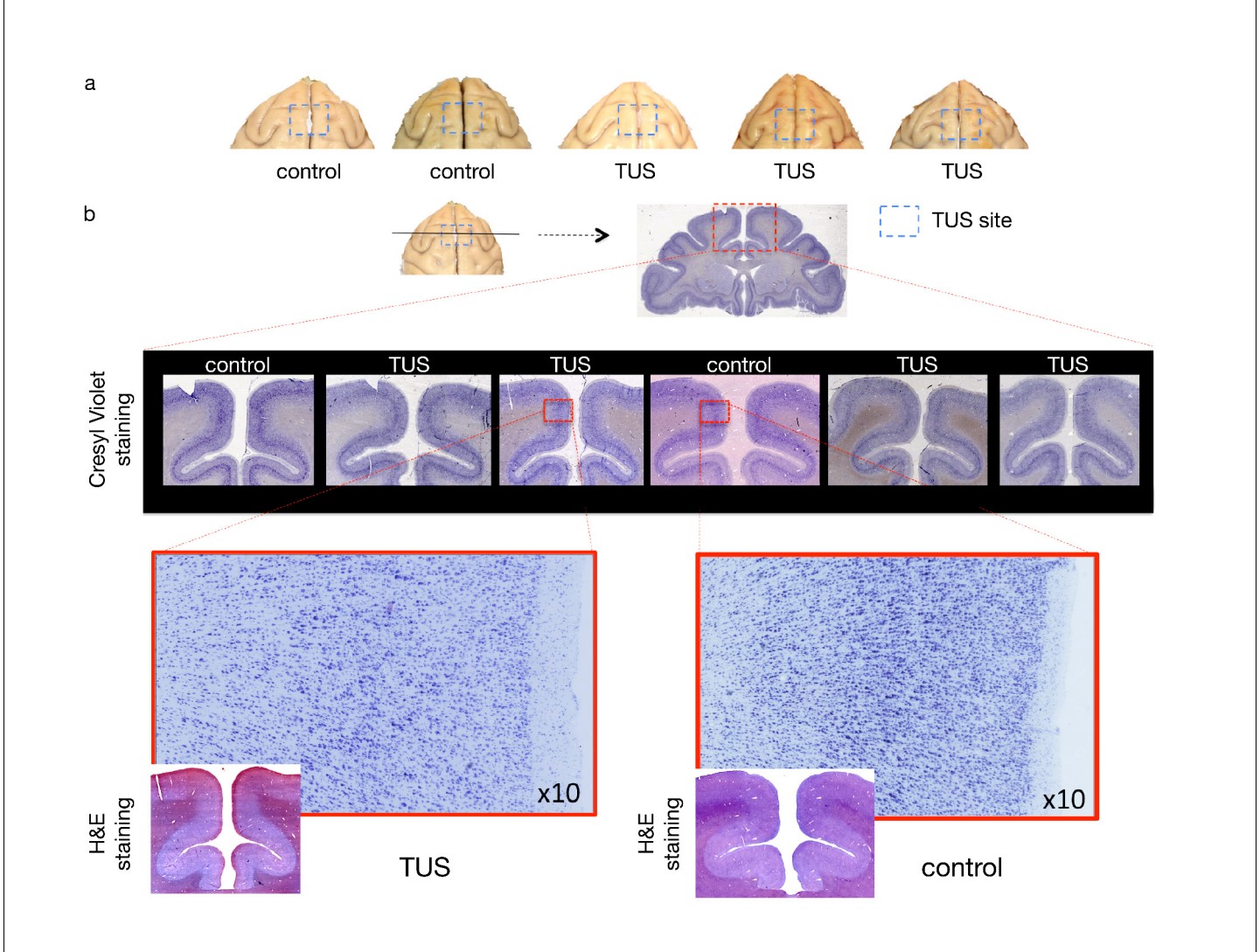

**Figure 8.** No effect of TUS was apparent on histological examination. (**a**) Dorsal view of perfused macaque brain. A post-mortem examination of the brain did not reveal macroscopic damage to the brain. (**b**) Histological assessment of 50-micron thick sections obtained at the level of the stimulation site with Cresyl Violet Nissl staining and H and E staining did not reveal any evidence of thermal lesions or haemorrhage after the TUS protocol used here. Magnified images are centred at the focal point of the stimulation.
DOI: https://doi.org/10.7554/eLife.40541.013

## Discussion

In this study we demonstrate a protocol for transcranial focused ultrasound stimulation (TUS) that can induce a sustained yet reversible change in neural activity. We focused our investigation on modulations of brain connectivity, following the notion that each brain area's unique contribution to cognition and behaviour is shaped by how activity in each area is a function of a unique fingerprint of interconnected areas (*Passingham et al., 2002*). We found that each area's connectional fingerprint was significantly changed by TUS, but only when it was applied to that area itself (*Figures 3,5*). The changes observed might be summarized as more uniform activation in the stimulated region combined with a sharpening of the normal coupling pattern that each area has even at rest. Activity coupling with strongly interconnected areas, which are often nearby, was increased but activity coupling with less strongly connected regions was reduced. Such changes in connectional fingerprints might constitute the mechanism by which TUS is able to induce regionally specific patterns of behavioural change when applied in awake behaving animals (*Deffieux et al., 2013*; *Fouragnan et al., 2019*).

The pattern of inputs each area receives from other areas and the influence it wields over other areas are a major determinant of its function and here we have shown that this pattern is altered by TUS. TUS may therefore provide a relatively straightforward method for sustained but reversible manipulation of specific components of neural circuits in the primate brain (*Wattiez et al., 2017*). This may be important for investigating primate brain areas when homologues in non-primate species, such as rodents, are non-existent or disputed (*Preuss, 1995*; *Wise, 2008*). This work paves the way for the development and use of offline TUS protocols in primates, including humans, both as a research tool and as potential clinical intervention.

In experiments 1 and 2, adopting a within-subject design with three animals, we found that TUS application produced different effects when applied to different brain regions: the SMA, a part of motor association cortex, and FPC a part of granular prefrontal cortex (*Figures 3,5*). However, in each case the TUS effects were prominent within the connectional fingerprint of the area stimulated. The connectional fingerprints of SMA and FPC are distinct (*Johansen-Berg et al., 2004*; *Neubert et al., 2014*; *Sallet et al., 2013*). The effects of TUS are thus regionally specific.

Our results confirm that TUS can be used as a neuromodulatory technique that allows one to non-surgically target cortical and subcortical brain areas with superior spatial specificity and depth of stimulation (*Folloni et al., 2019*) compared to other transcranial stimulation approaches (e.g. TMS and TCS; *Bestmann and Walsh, 2017*; *Dayan et al., 2013*; *Polanía et al., 2018*). While successes have been achieved with some invasive techniques, such as electrical microstimulation (*Krug et al., 2015*; *Vanduffel, 2016*), it is not easy to use them to disrupt activity in all areas especially when they are not somatotopically mapped. Recently it has been reported that some online TUS protocols in rodents induce neural changes as an indirect consequence of the auditory stimulation they entail (*Guo et al., 2018*; *Sato et al., 2018*). The spatially specific effects that we observed after TUS cannot, however, be attributed to any common auditory impact that occurs at the time of stimulation. Moreover, in order to avoid both the confounding effect associated with the sound of TUS and interference of the ultrasonic wave field with fMRI measurement, we opted for an 'offline' stimulation protocol. Stimulation was a 40 s train that ended at least 20 min before the fMRI data acquisition period. The sustained nature of the train and other features of the stimulation pulses may make the protocol used here more effective for neuromodulation, while ensuring the thermal modulation of the cortex remains limited (<1°C, *Figure 2*). Such limited thermal changes are not associated with neuromodulatory effects observed more than 30 min after the stimulation: the thermal rise is short-lived (*Dallapiazza et al., 2018*), not accompanied by tissue damage and below the thermal effects observed with some protocols in rodents (for a review, see *Constans et al., 2018*).

In experiment 3 we found that TUS had reproducible effects. When TUS was applied to FPC in three other individuals, it induced spatially specific effects similar to those seen in experiment 2.

The effects of the non-invasive 40 s stimulation protocol used here are sustained and lasted over much of the two-hour period we investigated. These effects are more extended than those produced by other techniques commonly used for offline disruption of cortical activity such as TMS (*Huang et al., 2005*; *O'Shea et al., 2007*). Care should therefore be taken in using the technique in human cognitive neuroscience experiments; TUS effects may continue beyond the short periods that participants typically spend within the laboratory. It may therefore be important to carefully characterize the time course of TUS effects in animal models before their use with human participants. Some caution might also be warranted in relation to the potential of ultrasound to cause microstructural damage, especially when stimulating at higher intensities, longer durations, or for more repetitions. While our structural MRI and histological analyses did not reveal any evidence of damage and were comparable to previous studies (*Dallapiazza et al., 2018*; *Lee et al., 2016b*), it would be of interest in the future to include additional histological indices of apoptotic or inflammatory processes (*Tufail et al., 2010*) to further assess the safety of TUS. Similarly, the thermal modelling approach we adopted here was designed to estimate an informed upper-bound on potential thermal effects, but when further developing this protocol additional simulation validations – for example based on phantom measurements – might be informative.

In addition to the spatially specific effects of TUS we also observed changes in the BOLD signal originating from the meningeal compartment that were not specific to the area stimulated. Although similar effects were seen each time either the SMA or FPC was stimulated, the effect was generally more pronounced after SMA stimulation than after FPC stimulation. Our preliminary results from TUS of other brain areas suggest that these spatially non-specific effects may be even smaller when

TUS is applied elsewhere (*Folloni et al., 2019*). While the precise origin of the non-specific effects was difficult to determine it is possible that they may result from a direct vascular effect of TUS; the sagittal sinus is directly above the midline frontal regions that we targeted and is included in the meningeal compartment during MRI analysis protocols. The presence of such non-specific effects again underlines the need for care in translating the technique to humans; especially when the targeted region is near venous sinuses or cerebral arteries. In addition, they underline the need for comparing the behavioural effects of TUS not just with a non-stimulation sham condition but with TUS application to another control brain region. The absence of marked histological changes in experiment 4, however, provides one important safety benchmark and confirms previous histological results in lagomorphs (*Yoo et al., 2011*).

Combining TUS and fMRI is a promising approach to overcome the restrictions of each of the individual techniques. Here we have shown that TUS has a detectable offline and sustained impact on the distinctive network of connectivity associated with the stimulated brain region – the connectional fingerprint. A brain region's interactions with other regions – its unique connectional fingerprint or specific pattern of inputs and outputs – are an important determinant of its functional role. The current results are therefore consistent with TUS application exerting regionally specific effects on behaviour (*Deffieux et al., 2013*; *Fouragnan et al., 2019*). The fact that fMRI allows the effects of TUS to be studied with a high spatial resolution suggests the TUS-fMRI combination has the potential to become a powerful neuroscientific tool.

# Materials and methods

## Subject details

For this study, six healthy male macaques (*Macaca mulatta*, NCBITaxon:9544) were stimulated with transcranial focused ultrasound and scanned to acquire resting state functional magnetic resonance images (rs-fMRI) and anatomical MR images. Three animals participated in experiment 1 [SMA TUS] (all males, mean age and weight at time of scan: 5.6 years, 10.7 kg). The same three animals participated in experiment 2 [FPC TUS] (at time of scan: 6.1 years, 11.8 kg), and the control condition (at time of scan: 5.5 years, 10.2 kg). Three different animals participated in experiment 3 [FPC TUS validation] (all males, mean age and weight at time of scan: 10.3 years, 13 kg). In addition to this set of animals, six animals were included in the histology analysis (experiment 4): three control animals who did not receive TUS (two females; mean age and weight at time of perfusion: 9.3 years, 9.1 kg) and three pre-SMA TUS animals post stimulation (all males; mean age and weight at time of perfusion: 8.4 years, 13.1 kg).

All procedures were conducted under licenses from the United Kingdom (UK) Home Office in accordance with The Animals (Scientific Procedures) Act 1986. In all cases they complied with the European Union guidelines (EU Directive 2010/63/EU).

## Ultrasound stimulation

A single element ultrasound transducer (H115-MR, diameter 64 mm, Sonic Concept, Bothell, WA, USA) with 51.74 mm focal depth was used with a coupling cone filled with degassed water and sealed with a latex membrane (Durex). The resonance frequency of the ultrasonic wave was set at 250 kHz with 30 ms bursts of ultrasound generated every 100 ms, controlled through a digital function generator (Handyscope HS5, TiePie engineering, Sneek, The Netherlands). The stimulation lasted for 40 s. A 75-Watt amplifier (75A250A, Amplifier Research, Souderton, PA) was used to deliver the required power to the transducer. A TiePie probe (Handyscope HS5, TiePie engineering, Sneek, The Netherlands) connected to an oscilloscope was used to monitor the voltage delivered. The recorded peak-to-peak voltage was kept constant throughout the stimulation. Voltage values per session ranged from 130 to 142 V, corresponding to 1.17 to 1.35 MPa as measured in water with an in house heterodyne interferometer (*Constans et al., 2017*). Based on numerical simulations (see *Acoustic and thermal modelling* below for more details), the maximum peak pressure ($P_{max}$) and $I_{sppa}$ at the acoustic focus point were estimated to be 0.88 MPa and 24.1 W/cm$^2$ for the SMA target, and 1.01 MPa and 31.7 W/cm$^2$ for the FPC target ($I_{spta}$: 7.2 W/cm$^2$ and 9.5 W/cm$^2$ for SMA and FPC, respectively). Each of the areas targeted in experiments 1–4 lie close to the midline. Therefore,

we applied a single train over the midline stimulating the target region in both hemispheres simultaneously.

In order to direct TUS to the target region, we guided the stimulation using a frameless stereo-taxic neuronavigation system (Rogue Research, Montreal, CA; RRID:SCR_009539) set up for each animal individually by registering a T1-weighted MR image to the animal's head. Positions of both the ultrasound transducer and the head of the animal were tracked continuously with infrared reflectors to inform online and accurate positioning of the transducer over the targeted brain region: SMA in experiment 1, (Montreal Neurological Institute (MNI) X, Y, and Z coordinates in mm [0.1 2 19]); FPC in experiment 2 [0.6 24 10]; FPC in experiment 3 [-0.7 24 11]; pre-SMA in experiment 4 [0.2 11 17]. The ultrasound transducer/coupling cone montage was placed directly onto previously shaved skin prepared with conductive gel (SignaGel Electrode; Parker Laboratories Inc.) to ensure ultrasonic coupling between the transducer and the animal's scalp. In the non-stimulation condition (control), all procedures (anaesthesia, pre-scan preparation, fMRI scan acquisition and timing), with the exception of actual TUS, matched the TUS sessions.

## Acoustic and thermal modelling

The acoustic wave propagation of our focused ultrasound protocol (at 130 V peak-to-peak voltage) was simulated using a k-space pseudospectral method-based solver, k-Wave (*Cox et al., 2007*) to obtain estimates for the pressure amplitude, peak intensity, spatial distribution, and thermal impact at steady state. 3D maps of the skull were extracted from a monkey CT scan (Kyoto University online database, ID 1478, 0.26 mm isotropic resolution). Soft tissues were assumed to be homogeneous, with acoustic values of water ($\rho_{tissue}$ =1000 kg/m$^3$ and $c_{tissue}$ =1500 m/s). In the bone, a linear relationship between the Hounsfield Units (HU) from the CT scan and the sound speed, as well as the density, was used. The power law model for attenuation is $\alpha_{att} = \alpha_1 * \phi^\beta$ where the porosity $\phi$ is defined by $\phi = \frac{\rho_{max} - \rho}{\rho_{max} - \rho_{water}}$ in the skull (*Aubry et al., 2003*). The attenuation coefficient for the acoustic propagation $\alpha_1$ depends on the frequency: $\alpha_1 = \alpha_0 f^b$. We set the parameters to $\rho_{max} = 2200 \text{ kg/m}^3$, $c_{max} = 3100 \text{ m/s}$, $\beta = 0.5$, $\alpha_0 = 8 \text{ dB/cm/MHz}^b$, $b = 1.1$ (*Constans et al., 2018*). The attenuation coefficient in bone accounts for both absorption and scattering.

The propagation simulation was performed at 250 kHz with a 150μs-long pulse signal (enough to reach a steady state). The transducer was modelled as a spherical section (63 mm radius of curvature and 64 mm active diameter). The simulated pulses were spatially apodized (r = 0.35) on the spherical section. Ultrasound propagates first through water before entering the skull cavity with the geometrical focal point located below the surface, inside the brain. Simulations were performed in free water, and the maximum amplitude obtained was used to rescale the results in skull (the transducer calibration indicates that the maximum amplitude in water at 130V is 1.2 MPa). The thermal modelling is based on the bio-heat equation (*Pennes, 1948*):

$$\rho C \frac{\partial T}{\partial t} = \kappa \nabla^2 T + q + w \rho_b C_b (T - T_a)$$

where $T$, $\rho$, $C$, $\kappa$ and $q$ are the temperature, density, specific heat, thermal conductivity and rate of heat production respectively. Heat production is defined as $q = \alpha_{abs} \frac{P2}{2\rho C}$ - 1, $\alpha_{abs}$ - 1 being the absorption coefficient and P the peak negative pressure. $\kappa$ is set to 0.528 W.m$^{-1}$.K$^{-1}$ in soft tissue and 0.4 W.m$^{-1}$.K$^{-1}$ in the skull; C is set to 3600 J.kg$^{-1}$.K$^{-1}$ in soft tissue and 1300 J.kg$^{-1}$.K$^{-1}$ in the skull (*Duck, 2013*). In the tissue, the absorption coefficient was set to $\alpha_{abs\ tissue} = 0.21 \text{ dB/cm/MHz}^b$ - 1 (*Goss et al., 1979*). In the skull the longitudinal absorption coefficient is proportional to the density with $\alpha_{abs\ max} = a_0/3 = 2.7 \text{dB/cm/MHz}^b$ - 1(*Pinton et al., 2012*). The last term corresponds to the perfusion process: $w$, $\rho_b$, $C_b$, and $T_a$ correspond to the blood perfusion rate, blood density, blood specific heat and blood ambient temperature respectively. These parameters are assumed homogeneous over the brain, although a more detailed description of the brain cooling processes can be found in the literature (*Wang et al., 2015*). The perfusion parameters are based on previous reports (*Pulkkinen et al., 2011*): $w$=0.008s$^{-1}$; $\rho_b$= 1030 kg.m$^{-3}$; $C_b$ = 3620 J.kg$^{-1}$.K$^{-1}$ and $T_a$ = 37°C.

The bioheat equation is solved by using a 3D finite-difference scheme in MATLAB (Mathworks, Natick, USA) with Dirichlet boundary conditions. Initial temperature conditions were 37°C in the

brain, skull and tissue, and 24°C in the water coupling cone. Simulations were run over 1 min pre-sonication, followed by 40 s of sonication, and 5 min post-sonication, closely following the experimental procedure.

## Macaque MRI acquisition

We acquired one MRI session per monkey per condition: in total we performed three sessions per animal across experiments 1 and 2, and one session per animal in experiment 3. The ultrasound sonication and subsequent MRI scans were performed under inhalational isoflurane gas anaesthesia using a protocol which has previously proven successful in preserving whole-brain functional connectivity as measured with BOLD signal (*Mars et al., 2013*; *Mars et al., 2011*; *Neubert et al., 2015*; *Neubert et al., 2014*; *O'Reilly et al., 2013*; *Sallet et al., 2013*; *Vincent et al., 2007*). In the case of the TUS conditions, fMRI data collection began only after completion of the TUS train. Anaesthesia was induced using intramuscular injection of ketamine (10 mg/kg), xylazine (0.125–0.25 mg/kg), and midazolam (0.1 mg/kg). Macaques also received injections of atropine (0.05 mg/kg, intramuscularly), meloxicam (0.2 mg/kg, intravenously), and ranitidine (0.05 mg/kg, intravenously). The anaesthetized animals were placed in an MRI-compatible frame (Crist Instruments) in a sphinx position and placed in a horizontal 3 T MRI scanner with a full-size bore. Scanning commenced ~2 hr after induction, when the clinical peak of ketamine had passed. Anaesthesia was maintained, in accordance with veterinary recommendation, using the lowest possible concentration of isoflurane to ensure that macaques were anaesthetized. The depth of anaesthesia was assessed and monitored using physiological parameters (heart rate and blood pressure, as well as clinical checks before the scan for muscle relaxation). During the acquisition of the functional data the expired isoflurane concentration was in the range 0.6–0.8%. Isoflurane was selected for the scans as it was previously demonstrated to preserve rs-fMRI networks (*Mars et al., 2013*; *Mars et al., 2011*; *Neubert et al., 2015*; *Neubert et al., 2014*; *O'Reilly et al., 2013*; *Sallet et al., 2013*; *Vincent et al., 2007*). Macaques were maintained with intermittent positive pressure ventilation to ensure a constant respiration rate during the functional scan, and respiration rate, inspired and expired $CO_2$, and inspired and expired isoflurane concentration were monitored and recorded using VitalMonitor software (Vetronic Services Ltd.). Core temperature and $SpO_2$ were also constantly monitored throughout the scan. A four-channel phased-array coil was used for data acquisition (Dr. H. Kolster, Windmiller Kolster Scientific, Fresno, CA, USA).

FMRI data were collected in experiments 1–3. In each session whole-brain BOLD fMRI data were collected for 3 runs of approximately 26 min each, using the following parameters: 36 axial slices; in-plane resolution, 2 × 2 mm; slice thickness, 2 mm; no slice gap; TR, 2000 ms; TE, 19 ms; 800 volumes per run. A minimum period of 10 days elapsed between sessions.

T1-weighted structural MRI scans were collected in experiments 1–3. A structural scan (average over up to three T1w images acquired in the same session) was acquired for each macaque in the same session, using a T1 weighted magnetization-prepared rapid-acquisition gradient echo sequence (0.5 × 0.5×0.5 mm voxel resolution).

## Macaque anatomical MRI pre-processing

The pre-processing and analysis of the MRI data was designed to follow the HCP Minimal Processing Pipeline (*Glasser et al., 2013*), using tools of FSL (https://fsl.fmrib.ox.ac.uk/fsl/fslwiki; RRID:SCR_002823), HCP Workbench (https://www.humanconnectome.org/software/connectome-workbench; RRID:SCR_008750), and the Magnetic Resonance Comparative Anatomy Toolbox (MrCat; https://github.com/neuroecology/MrCat; copy archived at https://github.com/elifesciences-publications/MrCat; *Verhagen, 2019*). The T1w images were processed in an iterative fashion cycling through brain-extraction, RF bias-field correction, and linear and non-linear template registration to the *Macaca mulatta* F99 atlas (*Van Essen and Dierker, 2007*). The initial skull stripping was performed using a multi-seeded implementation of BET (*Smith, 2002*) optimized for macaque brains, while subsequent brain extraction was based on a high-fidelity template registered to the F99 macaque space. The RF bias-field was estimated and corrected using a robust implementation of FAST (*Zhang et al., 2001*). Linear and non-linear registration to F99 space was achieved using FLIRT (*Jenkinson et al., 2002*; *Jenkinson and Smith, 2001*) and FNIRT (*Andersson et al., 2007*; *Jenkinson et al., 2012*) with configurations adjusted to reflect macaque rather than human brain

characteristics. The application of a robust and macaque-optimised version of FAST also resulted inestimated compartments for grey matter, white matter, and meninges with cerebral spinal fluid. For each compartment a posterior-probability map was created by integrating a set of prior probability maps based on 112 *Macaca mullata* individuals (*McLaren et al., 2009*) with the dataset-specific evidence provided by FAST. Segmentation of subcortical structures was achieved by registration to the D99 atlas (*Reveley et al., 2017*).

## Macaque rs-fMRI pre-processing

The first 5 volumes of the functional EPI datasets were discarded to ensure a steady RF excitation state. EPI timeseries were motion corrected using MCFLIRT. Given that the animals were anaesthetized and their heads were held in a steady position, any apparent image motion, if present at all, is caused by changes to the B0 field and temperature, rather than by head motion. Accordingly, the parameter estimates from MCFLIRT can be considered to be 'B0-confound parameters' instead. Each timeseries was checked rigorously for spikes and other artefacts, both visually and using automated algorithms; where applicable slices with spikes were linearly interpolated based on temporally neighbouring slices. Brain extraction, bias-correction, and registration was achieved for the functional EPI datasets in an iterative manner, similar to the pre-processing of the structural images with the only difference that the mean of each functional dataset was registered to its corresponding T1w image using rigid-body boundary-based registration (FLIRT). EPI signal noise was reduced both in the frequency and temporal domain. The functional timeseries were high-pass filtered with a frequency cut-off at 2000 s. Temporally cyclical noise, for example originating from the respiration apparatus, was removed using band-stop filters set dynamically to noise peaks in the frequency domain of the first three principal components of the timeseries.

To account for remaining global signal confounds we considered the signal timeseries in white matter (WM) and meningeal compartments. Specifically, the WM + meningeal confound timeseries was described by the mean time course and the first five subsequent principal components of the combined WM and meningeal compartment (considering only voxels with a high posterior probability of belonging to the WM or meningeal compartment, obtained in the T1w image using FAST). The principal components of the WM + meningeal signal were estimated using a singular value decomposition approach. The B0 confound parameter estimates obtained from MCFLIRT were expanded as a second degree Volterra series to capture both linear and non-linear B0 effects. Together the WM + meningeal and expanded B0 confound parameters were regressed out of the BOLD signal for each voxel.

In a separate analysis (*Figure 6* and *Figure 6—figure supplement 1*), to assess the contribution of the meningeal compartment signal we repeated the identical procedure as above, with the only difference that the mean and principal components were extracted from signal in the WM compartment alone, excluding the meningeal compartment.

Following this confound cleaning step, the timeseries were low-pass filtered with a cut-off at 10 s. The cleaned and filtered signal was projected from the conventional volumetric representation (2 mm voxels) to the F99 cortical surface (~1.4 mm spaced vertices), while maintaining the subcortical volumetric structures. The data were spatially smoothed using a 3 mm FWHM gaussian kernel, while considering the folding of the cortex and the anatomical boundaries of the subcortical structures. Lastly, the data timeseries were demeaned to prepare for functional connectivity analyses.

## Macaque rs-fMRI analyses

To represent subject effects, the timeseries from the three runs were concatenated to create a single timeseries of $3 \times (800 - 5) = 2385$ volumes per animal per intervention (control, SMA TUS, FPC TUS). To represent group effects the run-concatenated timeseries of all animals were combined using a group-PCA approach resulting in a series of 200 volumes representing the principal components (Smith et al., 2014).

We report on the whole-brain functional coupling of the stimulation sites in the control state and after TUS, adopting a seed-based correlation analysis approach. Effects of TUS are quantified based on a limited set of regions, a 'fingerprint', whose strength of interconnection with macaque SMA or FPC is known from anatomical tracing studies (*Bates and Goldman-Rakic, 1993*; *Dum and Strick, 2005*; *Geyer et al., 2000*; *Lu et al., 1994*; *Petrides and Pandya, 2007*; *Strick et al., 1998*) and

from studies of fMRI activity coupling under anaesthesia (*Neubert et al., 2015*; *Neubert et al., 2014*; *Sallet et al., 2013*). For example, SMA is strongly interconnected with core nodes in the sensorimotor network, including M1, SPL, and MCC. The unique profile of interconnections characterizing a functional area is defined by a pattern of both strong and limited interconnections (*Passingham et al., 2002*). Accordingly, we selected a set of regions based on either distinctively strong or distinctively limited interconnection and fMRI coupling with SMA or FPC. In fact, as a result of the complementary patterns of connections defining SMA and FPC (*Petrides and Pandya, 2007*), areas that were included based on their known strong connection and coupling with FPC (9m, 9-46d, PCC, IPLc, midSTS) were also included because of their known limited connections and coupling with SMA. Furthermore, to characterize the effect of TUS targeted at FPC on adjacent prefrontal cortex, we also included additional ventromedial and orbital prefrontal regions with which FPC shared less strong connections. To ensure a balanced distribution of fingerprint targets across the brain and to ensure that both SMA and FPC fingerprints were comprised of the same number of regions we consider areas 11m and aSTG in the SMA fingerprint, both regions with which the SMA is weakly interconnected. Lastly, to test for specificity of the TUS effects, we included SMA as a target for the FPC fingerprint and FPC as a target for the SMA fingerprint.

To construct regions-of-interest (ROI) for SMA and FPC, circles of 4 mm radius were drawn on the cortical surface around the point closest to the average stimulation coordinate (*Figure 2*), in both the left and the right hemisphere. The same procedure was used to define other bilateral cortical regions of interest, based on literature coordinates (*Mars et al., 2011*; *Neubert et al., 2015*; *Neubert et al., 2014*; *Sallet et al., 2013*), to serve as seeds for connectivity analyses (A1, *Figure 5g–i*; POp, *Figure 6*, *Figure 6—figure supplement 1*) or targets for the fingerprint analyses (*Figure 5*). Apart from the stimulation sites and primary auditory cortex, all ROIs were selected based on their known anatomical connectivity, of relevance because they are known to be distinctively strongly or distinctively weakly connected with the stimulation sites.

Coupling between the activity of each region of interest and the rest of the brain was estimated by calculating the correlation coefficient between each point in the ROI and all other data points (Fisher's $z$: inverse hyperbolic tangent of Pearson's $r$, bounded at $[-2\ 2]$). The resulting 'connectivity-maps' were averaged across all vertices/voxels in the ROI, and subsequently averaged across hemispheres. Accordingly, the final maps represent the average coupling of a bilateral ROI with the rest of the brain. The fingerprints are obtained by extracting the average coupling with each target ROI and averaging across the two hemispheres.

The strength of self-connections within SMA and FPC ROIs was estimated in the same way as the coupling of either SMA or FPC with any remote ROI was determined. Namely, for each point in the ROI the Fisher's $z$-transformed correlation between this point's timeseries and that of each of the other points in the ROI was calculated. Subsequently, for each point the resultant $z$-values were averaged, describing local coupling at that point, and averaged across the whole ROI to obtain a single estimate of self-coupling per ROI (*Figure 5c*).

To assess the contribution of the meningeal compartment signal, we quantified the effect of TUS for each point on the cortical surface in data that was not corrected for the meningeal signal (see *Macaque rs-fMRI pre-processing*). For each point, we calculated it's coupling with the rest of the brain, sorted all Fisher's $z$-values in ascending order, and extracted the $z$-value at the 98[th]-percentile. $Z$-values at this level describe relatively strong coupling with the seed point and are often observed in the vicinity of the seed region or other strongly connected regions (depicted in bright yellow in *Figure 6*), ensuring that irrelevant connections are ignored. As such, this simple statistic is well-suited to capture any main effects of TUS on coupling strength, including those observed when comparing panels (b) and (e) in *Figure 6*. This statistic allows us to quantify the effect of TUS in a single value for each point on the cortical surface, and to directly contrast the values obtained in the control state with those obtained after TUS (*Figure 6f*, *Figure 6—figure supplement 1f,i*).

We described the signal variance in the GM and meningeal compartments, or more specifically, the variance explained by the first five principal components (*Figure 6c*, *Figure 6—figure supplement 1* panel c), using the same approach that was used to decompose signal in WM or WM + meningeal compartments in the EPI timeseries cleaning step, as described above. We defined the explained variance as the sum of the first five eigenvalues of the covariance matrix of the compartment signals divided by the sum of all eigenvalues.

## Statistics

All suitable animals available at the time of experimentation took part in this study. Accordingly, there was no pre-selection nor restriction for group allocation. No data was excluded for analysis. Sample sizes could not be predetermined statistically in the absence of a prior literature reporting relevant expected effect sizes; instead we adopted sample sizes similar to those reported in previous publications detailing (interventional) macaque fMRI studies (*Chau et al., 2015*; *O'Reilly et al., 2013*; *Papageorgiou et al., 2017*). Data collection and analysis were not performed blind to the conditions of the experiments. We used a group PCA approach to define the centre of group rs-fMRI effects (see *Macaque rs-fMRI analyses*). For all other effects the centre of the group is defined as the mean of the individual subject effects.

To report on the full extent of the TUS effects and the simple effects that drive any observed differences we reproduce the whole-brain functional connectivity maps unthresholded for the control, SMA TUS, and FPC TUS conditions separately. In order to make a statistical comparison of the functional coupling of the SMA and FPC in the control and TUS conditions it is problematic to compare coupling at each and every other point in the brain because there is a risk of false positive effects if multiple comparisons are made. Given the limited sample sizes possible with non-human primate experiments, however, there is a risk of false negative results if stringent correction for multiple comparisons is undertaken at the whole-brain level. Indeed, here we avoid these pitfalls and make inference on a single statistic that combines information from a planned limited set of regions defined by coordinates from previous studies (*Mars et al., 2011*; *Neubert et al., 2015*; *Neubert et al., 2014*; *Sallet et al., 2013*).

Importantly, rather than examining activity coupling between the seed area and each of the fingerprint ROIs in turn and risking potential false positive results, we compared the overall pattern of coupling using the method devised by Mars and colleagues (*Mars et al., 2016*); non-parametric permutation tests were performed on cosine similarity metrics summarizing pairs of fingerprints (SMA TUS versus control and FPC TUS versus control). Each connectional fingerprint can be represented as a multidimensional vector, with the number of dimensions corresponding to the number of target ROIs in the fingerprint. The cosine similarity metric quantifies the angle between two of such multi-dimensional vectors. As such, it takes into account the shape of the fingerprint irrespective of its mean amplitude and results in a single metric per pair of fingerprints, negating the necessity for correcting for multiple comparisons across fingerprint targets. To assess the likelihood of observing this cosine similarity metric under the null-hypothesis of no difference between conditions, it needs to be compared against a distribution of metrics obtained from a random sample. To avoid non-compliance to the requirements for a conventional parametric approach (e.g. independence of sampling and adherence to a Gaussian normal distribution) we applied a non-parametric permutation approach, in which the parent distribution is empirically drawn by repeatedly (randomly) shuffling the group-membership of data points and re-calculating the metric based on the shuffled labelling (e.g. data points acquired in the control state might be randomly assigned to the intervention condition, and vice versa). In this way the empirical distribution under the null-hypothesis does not rely on assumptions about the shape of the distribution but will acknowledge dependencies between target ROIs in the fingerprint; as such this approach will avoid inflation of type I error (*Maris and Oostenveld, 2007*; *Winkler et al., 2014*). The relatively small number of TUS sites, animals, and fMRI runs allowed us to exhaustively test all possible permutations (24309) to obtain the true probability of rejecting the null hypothesis.

All other statistical inferences were drawn in the context of generalized linear mixed-effects (GLME) models that considered the intercept and factorial design (including interactions where appropriate) as fixed effects and the intercept and slope grouped per animal as random effects with possible correlation between them (as implemented in MATLAB, Mathworks, Natick, USA). The models were assumed to adhere to a normal distribution of the data (not formally tested) and were fitted using Maximum-Pseudo-Likelihood estimation methods where the covariance of the random effects was approximated using Cholesky parameterization. Statistical significance was set at $\alpha$ = 0.05, two-tailed, and estimated using conventional analyses of variance (ANOVA). For all parametric tests we report the test statistic ($F$), probability estimate ($p$), effect size (Cohen's $d$), and the lower and upper limits of the 95% confidence interval (CI). For all plots, the central tendency across individual animals is derived directly from the group-PCA approach(*Smith et al., 2014*) or

described as the mean across all animals (n = 3). Dispersion is described as the standard-error of the mean (SE).

## Histology

In experiment 4, prior to histological examination, animals were anaesthetized with sodium pentorbarbitone and perfused with 90% saline and 10% formalin. A post-mortem examination of the surface of the brain was conducted prior to the brain extraction. The brains were then removed and placed in 10% sucrose formalin. The brains were blocked in the coronal plane at the level of the lunate sulcus. Each brain was cut in 50 µm coronal sections. Every tenth section, two adjacent sections were retained for analysis whereby one was stained with Cresyl Violet (Nissl body staining) and the other with Haemotoxylin and Eosin (H and E staining), in line with previous studies (*Dallapiazza et al., 2018*; *Lee et al., 2016b*).

## Acknowledgements

We would like to thank Greg Daubney for his assistance with the histological preparation and analysis, and Dr. Constantin Coussios and Dr. Robin Cleveland for their technical support. Funding for this research was provided by a Wellcome Trust UK Henry Dale fellowship (105651/Z/14/Z), Wellcome Trust Senior Investigator Award (WT100973AIA), Medical Research Council UK (MRC) programme grant (MR/P024955/1), MRC programme grant (G0902373), Wellcome Trust UK Grant (105238/Z/14/Z), Wellcome Trust Sir Henry Wellcome fellowship (103184/Z/13/Z), Biotechnology and Biological Sciences Research Council UK (BBSRC) grant (BB/N019814/1), The Netherlands Organisation for Scientific Research (NWO) grant (452-13-015), The Wellcome Centre for Integrative Neuroimaging (203139/Z/16/Z), Bettencourt Schueller Foundation, and the Agence Nationale de la Recherche under the Future Investments program (ANR-10-EQPX-15).

## Additional information

### Funding

| Funder | Grant reference number | Author |
|--------|------------------------|--------|
| Wellcome | 203139/Z/16/Z | Lennart Verhagen<br>Davide Folloni<br>Daria EA Jensen<br>Léa Roumazeilles<br>Miriam C Klein-Flügge<br>Rogier B Mars<br>Matthew FS Rushworth<br>Jerome Sallet |
| Wellcome | WT100973AIA | Lennart Verhagen<br>Miriam C Klein-Flügge<br>Matthew FS Rushworth |
| Bettencourt Schueller Foundation | | Cécile Gallea<br>Charlotte Constans<br>Harry Ahnine<br>Mathieu Santin<br>Stéphane Lehericy<br>Pierre Pouget<br>Jean-François Aubry |
| Agence Nationale de la Recherche | ANR-10-EQPX-15 | Cécile Gallea<br>Charlotte Constans<br>Harry Ahnine<br>Mathieu Santin<br>Stéphane Lehericy<br>Pierre Pouget<br>Jean-François Aubry |
| Wellcome | 105238/Z/14/Z | Davide Folloni |

| Medical Research Council | MR/P024955/1 | Davide Folloni<br>Miriam C Klein-Flügge<br>Matthew FS Rushworth<br>Jerome Sallet |
| --- | --- | --- |
| Wellcome | 103184/Z/13/Z | Miriam C Klein-Flügge |
| Biotechnology and Biological Sciences Research Council | BB/N019814/1 | Rogier B Mars |
| Nederlandse Organisatie voor Wetenschappelijk Onderzoek | 452-13-015 | Rogier B Mars |
| Medical Research Council | G0902373 | Matthew FS Rushworth<br>Jerome Sallet |
| Wellcome | 105651/Z/14/Z | Jerome Sallet |

The funders had no role in study design, data collection and interpretation, or the decision to submit the work for publication.

## Author contributions

Lennart Verhagen, Resources, Data curation, Software, Formal analysis, Supervision, Validation, Investigation, Visualization, Methodology, Writing—original draft, Writing—review and editing; Cécile Gallea, Formal analysis, Investigation, Methodology, Writing—review and editing; Davide Folloni, Data curation, Formal analysis, Investigation, Methodology, Writing—review and editing; Charlotte Constans, Data curation, Software, Formal analysis, Investigation, Methodology, Writing—review and editing; Daria EA Jensen, Data curation, Formal analysis, Writing—review and editing; Harry Ahnine, Léa Roumazeilles, Mathieu Santin, Bashir Ahmed, Investigation, Writing—review and editing; Stéphane Lehericy, Resources, Writing—review and editing; Miriam C Klein-Flügge, Rogier B Mars, Software, Methodology, Writing—review and editing; Kristine Krug, Resources, Methodology, Writing—review and editing; Matthew FS Rushworth, Resources, Formal analysis, Supervision, Funding acquisition, Investigation, Methodology, Writing—original draft, Project administration, Writing—review and editing; Pierre Pouget, Conceptualization, Resources, Supervision, Investigation, Methodology, Project administration, Writing—review and editing; Jean-François Aubry, Conceptualization, Resources, Formal analysis, Supervision, Funding acquisition, Investigation, Methodology, Project administration, Writing—review and editing; Jerome Sallet, Conceptualization, Resources, Data curation, Formal analysis, Supervision, Funding acquisition, Investigation, Methodology, Project administration, Writing—review and editing

## Author ORCIDs

Lennart Verhagen (iD) http://orcid.org/0000-0003-3207-7929

Davide Folloni (iD) http://orcid.org/0000-0003-1653-5969

Charlotte Constans (iD) http://orcid.org/0000-0001-6378-9158

Miriam C Klein-Flügge (iD) http://orcid.org/0000-0002-5156-9833

Kristine Krug (iD) http://orcid.org/0000-0001-7119-9350

Pierre Pouget (iD) http://orcid.org/0000-0002-4721-7376

Jean-François Aubry (iD) http://orcid.org/0000-0003-2644-3945

Jerome Sallet (iD) http://orcid.org/0000-0002-7878-0209

## Ethics

Animal experimentation: All procedures were conducted under project licenses from the United Kingdom (UK) Home Office in accordance with The Animals (Scientific Procedures) Act 1986. In all cases they complied with the European Union guidelines (EU Directive 2010/63/EU).

## Decision letter and Author response

Decision letter https://doi.org/10.7554/eLife.40541.018

Author response https://doi.org/10.7554/eLife.40541.019

## Additional files

### Supplementary files
• Transparent reporting form
DOI: https://doi.org/10.7554/eLife.40541.014

### Data availability

FSL can be downloaded from https://fsl.fmrib.ox.ac.uk/fsl/fslwiki. HCP Workbench can be downloaded from https://www. humanconnectome.org/software/connectome-workbench. For any information regarding MrCat please see http://www.rbmars.dds.nl/lab/toolbox.html; further inquiries can be directed to Lennart Verhagen (lennart.verhagen@psy.ox.ac.uk). All dedicated software tools are available at https://github.com/neuroecology/MrCat. Data contributing to this work have been publicly deposited at the Wellcome Centre for Integrative Neuroimaging (https://git.fmrib.ox.ac.uk/lverhagen/offlinetus). For any inquiries regarding the data, please contact Jérôme Sallet (jerome. sallet@psy.ox.ac.uk).

The following dataset was generated:

| Author(s) | Year | Dataset title | Dataset URL | Database and Identifier |
|---|---|---|---|---|
| Lennart Verhagen | 2019 | offlineTUS | https://git.fmrib.ox.ac. uk/lverhagen/offlinetus | Wellcome Centre for Integrative Neuroimaging, offlinetus |

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
