## [Decision Letter]

[Editors’ note: the authors were asked to provide a plan for revisions before the editors issued a final decision. What follows is the editors’ letter requesting such plan.]

Thank you for sending your article entitled "Offline impact of transcranial focused ultrasound on cortical activation in primates" for peer review at *eLife*. Your article is being evaluated by three peer reviewers, and the evaluation is being overseen by a Reviewing Editor and Joshua Gold as the Senior Editor.

Given the list of essential revisions, including new experiments, the editors and reviewers invite you to respond within the next two weeks with an action plan and timetable for the completion of the additional work. We plan to share your responses with the reviewers and then issue a binding recommendation.

The consensus was clear that this is potentially important work, very well conducted overall and potentially of high impact to the field of non-invasive brain stimulation. However, there were some potentially more serious concerns, which are detailed below. Therefore, before making the final recommendation, we suggest possible actions that you could take to remediate these concerns.

While this seems an unusual step, *eLife* fortunately allows for such a more informal route. We think this will help to discuss a realistic plan for a revision, and to clarify any misunderstandings prior to making an editorial decision (wrt the points below).

Please note that while we outline possible actions below, there may well be other ways to address these concerns, hopefully within a realistic timeline. We wouldn't sweat over the usual two months timeline should you feel that more time was needed.

Comments:

If our understanding is correct, then the main concern relates to the experimental design and disambiguating the effect of TUS with inter-animal differences in functional connectivity.

The comparison of brain connectivity of the stimulated animals with the brain connectivity of other non-stimulated animals raises concern about the contribution of inter-individual variability in the connectivity as a potential confound to the findings. Differences between stimulated and non-stimulated animals would seem to reveal variability comprising both the effect of TUS as well as any across-individual differences. The concern is that the latter can be substantial. Related, in the case of the TUS conditions, fMRI data collection began only after completion of the TUS train.

Specifically, our concern arises from the description of the procedures, stating that 'three animals participated in experiments 1 and 2, including the control condition (all males, mean age and weight at time of scan: control, 5.5 years, 10.2 kg; SMA TUS, 5.6 years, 10.7 kg; FPC TUS, 6.1 years, 11.8 kg).' This implies that each animal took part in one condition. Yet it is also stated that the same animals 'took part in experiments 1 and 2', and 'The same three animals participated in experiments 1 and 2, and a control experiment conducted in the absence of TUS.' At the same time, it is stated that 'while experiments 1 and 2 followed a within-subject design, for experiment 3 we conducted a between-subject analysis where the subjects in the two experiments differed in age.' – this would imply that in E1 and E2 a within-subject comparison was done (though it was not clear whether the control comparison was within or between animals). If you could provide a clearer description of the exact procedures and comparisons, this would greatly help with the discussion.

Therefore, to which degree could the results arise from significant across-individual differences (as opposed to TUS + across-subject variability)? One observation that may point to this being at least a possibility is that the connectivity of the stimulated animals does not seem to be converging to that of the non-stimulated animals even after 1.5 hours.

It would seem that a within-subject comparison of the functional connectivity before, during (if possible) and after TUS is necessary. This procedure could then be repeated for non-stimulated animals to control for the effect of time on connectivity, i.e. compare pre-TUS FC maps for each animal and then post-TUS or post control period maps, for the two target sites. This would address whether the effects that were observed were significantly larger than those that would be expected from inter-animal differences.

However, we are aware that repeating the study in full with a within-subject design might be unfeasible, and would therefore discuss a realistic way forward given everyone agreed about the relevance of your work and its likely impact. We would not necessarily expect another large dataset, and a small set of animals may be able to provide reassurance regarding the point above.

And there may also be other ways to address the concerns. Would a 'normative' dataset, to which your group might have access to, provide an alternative route, e.g. resting state datasets from other monkey MRI projects? We discussed whether comparison of your animals with such a dataset potentially address the concerns, e.g. by showing how each animal differs from such a normative dataset? As mentioned above, we're happy to hear suggestions given the more substantial expertise on these type of data in your group.

There would be other comments should we proceed, but we first wanted to hear your thoughts and how you may be able to address the concerns.

[Editors’ note: formal revisions were requested, following approval of the authors’ plan of action.]

Thank you for sending your article entitled "Offline impact of transcranial focused ultrasound on cortical activation in primates" for peer review at *eLife*. Your article is being evaluated by three peer reviewers, and the evaluation is being overseen by a Reviewing Editor and Joshua Gold as the Senior Editor. Two of the reviewers have agreed to share their identity: Stefan Everling (Reviewer 1) and Jacek Dmochowski (Reviewer 3).

Thanks for the quick reply, which has greatly clarified things. We're happy to invite a revision based on these, and are confident you can provide these within the usual timeframe.

Perhaps most critically then would be to clarify the procedures throughout (after all, five people could not confidently work this out), leaving no ambiguity to exactly what was done. The key point of reviewer 3 (and related reviewer 1 and the two editors) can otherwise be ignored. Elsewhere, I am confident that you will find the requested revisions are straightforward to address.

Other points you should address:

In addition, we'd like you to provide additional detail on the procedures, as raised by reviewer 3.

Effect sizes remained a concern (see comments by reviewer 1 and reviewer 3). These may be resolved by revising the figures, but addressing the concern about the procedures for determining and analysing connectivity fingerprints would be helpful. Please also clarify the concern regarding the reporting of the statistical procedures.

Some additional detail on exactly how the fingerprints of stimulated animals were compared to non-stimulated animals (i.e., how statistical significance of the fingerprints was computed) would be helpful. For example, the Materials and methods section merely refers to a paper (Mars et al., 2016), which refers to another paper (Nichols and Holmes, 2007) on the important issue of how the permutations were performed.

For some analyses multiple tests were conducted. What procedures for corrections for multiple comparisons were in place throughout?

The sizes of the correlation vectors being compared do not seem to be provided either in the text or figure captions, but would be useful to see.

The choice of frontal polar cortex seems valid as a control because it has a very distinct functional connectivity from SMA. Is the SNR comparable for these two regions? To rule out the possibility that differences between FPC and SMA are simply related to SNR differences, could you should show temporal SNR maps and quantify these differences, if any?

It is virtually impossible to appreciate the differences in the FC maps in Figure 2. The addition of difference maps might help here (e.g. TUA over SMA minus control).

Regarding the acoustic parameter, 30 ms burst in every 100 ms (i.e. pulse repetition of 10 Hz; i.e. duty cycle of 30%), seems unconventional, even though they appear to be effective here. 10 Hz is well below the audible frequency, which helped to disregard effects from auditory involvement. However, an additional rationale for the choice of parameter (or brief summary of the literature) will be helpful.

With regard to the acoustic and Thermal modelling, the conservative approach taken here may over-estimate the actual temperature rise in the skull and the brain. Although additional experiments are not necessary, realistic estimation of temperature rise using phantoms would be desirable. If access to such data exist we would encourage you to include these.

In the histological analysis, a rather limited staining approach is used (i.e. Cresyl Violet and H&E) to evaluate the sonicated brain tissue. Although not absolutely needed in present study, there are other types of histological analyses to assess the presence of tissue damages (such as apoptotic activities or ischemic damages). At the very least, please reflect on these limitations in the Discussion.

Transcranial focused ultrasound was abridged to 'TUS'. As the method utilized the focused version (i.e. FUS), the change in acronym to tFUS might be more intuitive.

Inclusion of some relevant published work may be helpful, at your discretion: (1) fMRI-tFUS human study(ies) and (2) recent findings regarding the presence of long-term neuromodulatory effects of tFUS in rodents.

Reviewer 1:

Verhagen and colleague used resting-state fMRI in macaque monkeys to investigate the effects of transcranial focused ultrasound (FUS) on functional brain connectivity. The study appears to be carefully conducted and it reveals some important aspects of FUS. First it shows that FUS increases local functional connectivity and mainly decreases distant functional connectivity. Second it demonstrates that the effects last for at least 2 hours. Overall the effects appear to quite small but they seem to be consistent across animals (Fg. 3c). I have a few major points that need to be addressed.

1) I assume the authors picked frontal polar cortex because it has a very distinct functional connectivity from SMA. Based on my experience, the SNR can be quite low for this anterior area. To rule out the possibility that differences between FPC and SMA are simply related to SNR differences, the authors should show temporal SNR maps.

2) I found it virtually impossible to see any differences in the FC maps in Figure 2. The addition of difference maps might help here (e.g. TUA over SMA minus control).

3) It is not clear from the paper how often each animal was actually scanned. Did the authors only perform one SMA and one FPC stimulation session per monkey or did they conduct multiple sessions?

Reviewer 2:

Verhagan and colleagues reported neuromodulatory effects that are present beyond the transcranial FUS (tFUS)-mediated brain stimulation in supplementary motor area and frontal polar cortex using functional MRI on non-human primate (NHP) model. This is the first study reported in NHP that shows the presence of long-term, neuromodulatory effects conferred by tFUS. In addition, the study found that the effects are entirely independent from the confounding effects from potential acoustic startle response (ARS), which is allegedly believed to be a part of the ultrasound-mediated brain stimulation (at least in rodent studies). Due to these two major findings, along with very robust and rigorous methods used in this study (including the PCA analysis regarding the signals originating from the meningeal compartment), the data and their interpretation are convincing. Publication of this report would be timely for scientific community and will provide new perspectives regarding the acoustic neuromodulation. This study also provides strong early evidence that the tFUS, given at the level, did not induce apparent damages to the brain tissue. The report deserves expedited publication.

Here are a few suggestions, with moderate importance, which may improve the manuscript.

1) Regarding the acoustic parameter, 30 ms burst in every 100 ms (i.e. pulse repetition of 10 Hz; i.e. duty cycle of 30%), is bit unconventional, but appeared to be effective. 10 Hz is well below the audible frequency, which helped the authors to disregard effects from auditory involvement. Additional rationale for the choice of parameter or brief review on literatures on the parameters in the Discussion section will be helpful.

2) In regard to the acoustic and Thermal modeling, the study seems to take conservative approach, which may over-estimate the actual temperature rise in the skull and the brain. Although additional experiments are not necessary, realistic estimation of temperature rise using phantoms will help to ascertain the non-thermal effects of tFUS.

3) In histological analysis, authors performed a rather limited staining (i.e. Cresyl Violet and H&E) to evaluate the sonicated brain tissue. Although not absolutely needed in present study, there are other types of histological analyses to assess the presence of tissue damages (such as apoptotic activities or ischemic damages), and authors are advised to reflect these limitations in the Discussion.

4) Transcranial focused ultrasound's acronym was defined as 'TUS'. As the method utilized the focused version (i.e. FUS), the change in acronym to tFUS is strongly suggested.

5) Although the detailed references are given in the technical methods section, surprisingly, only a few previous investigations on the matter of main subjects (fMRI and long-term effects of tFUS) are given. Inclusion of (1) fMRI-tFUS study(ies) among humans and (2) recent findings regarding the presence of long-term neuromodulatory effects of tFUS on rodents (a simple Pubmed search will do) is advised for general audience.

Reviewer 3:

Summary:

In this paper the authors investigate a relatively new form of neuromodulation using ultrasound (Transcranial Ultrasound Stimulation or TUS). TUS was applied over the supplementary motor area (SMA) and frontal polar cortex (FPC) in primates, and resting-state functional connectivity was assessed at least 20 minutes following the stimulation. Control animals were scanned but without first receiving TUS. The authors argue that the "connectivity fingerprint", a measure of the connection strength between the stimulated area (SMA or FPC) and other brain regions was altered by TUS. In particular, they argue that regions with which the seed connects to strongly have their connections increased, while regions weakly connected are further weakened. They perform a replication study and also argue for changes in the meningeal (CSF) compartments of the brain.

Major concerns:

Inadequacy of control condition. In my view, the paper suffers from a suboptimally chosen control. Namely, the authors compare the brain connectivity of the stimulated animals with the brain connectivity of other non-stimulated animals. This selection means that inter-individual variability in the connectivity is not controlled for and has the potential to confound the findings. Given two animals A and B, animal A receives stimulation and animal B does not. Comparing A and B on any outcome measure leads to total variability comprising both the effect of TUS as well as any across-individual differences (sorry if I am stating what is obvious, I just want to be clear). The nature of functional connectivity and its shaping by experience could seemingly lead to significant across-individual differences which have not been examined. The authors have not provided data arguing that the observed differences (i.e., from TUS + across-subject variability) are significantly larger than those from across-individual variability alone. One observation that emphasizes this concern is that the connectivity of the stimulated animals does not seem to be converging to that of the non-stimulated animals even after 1.5 hours (see Figure 3F, where the difference between red and blue over M1-SPL-MCC is actually larger than at 1 hour).

Lack of rigour in statistical procedures. There is a significant absence of necessary detail regarding how exactly the fingerprints of stimulated animals were compared to non-stimulated animals (i.e., how statistical significance of the fingerprints was computed). The Materials and methods section refers to a paper (Mars et al., 2016), which refers to another paper (Nichols and Holmes, 2007) on the important issue of how the permutations were performed. Another issue is the apparent lack of correction for multiple comparisons, even though it seems that multiple tests were conducted. The sizes of the correlation vectors being compared are not provided either in the text or figure captions. I also have concerns about how the areas that comprise the fingerprint of a given region were selected, and how this selection influences the results of the statistical tests.

Magnitude of the effect on connectivity fingerprint does not appear to be very large. I list this as a major concern because *eLife* aims to publish research of the "highest scientific standards and importance." Combined with the issues described above, it is my opinion that, taken as a whole, the data is not convincing enough to justify the authors' conclusions that TUS altered functional connectivity in the macaque brain following stimulation.

I believe that a more suitable approach would have been to compare functional connectivity before, during (if possible) and after TUS within an animal. This procedure would then be repeated for non-stimulated animals to control for the effect of time on connectivity. It would also have been more convincing to observe effects when comparing the entire functional connectivity matrices (instead of just the pruned fingerprints, especially since little information was provided as to how the pruning was achieved).

In summary, I commend the authors for undertaking what were clearly technically challenging and involved experiments. They have also done a good job in communicating the results and preparing nice figures. While I believe that TUS has promise as a technique for a neuromodulation, I do not feel that the data presented here are strong enough to warrant publication in *eLife*. I am sorry that I cannot be more positive in my review.

---

## [Author Response]

[Editors’ note: what follows is the authors’ plan to address the revisions.]

Comments:If our understanding is correct, then the main concern relates to the experimental design and disambiguating the effect of TUS with inter-animal differences in functional connectivity.The comparison of brain connectivity of the stimulated animals with the brain connectivity of other non-stimulated animals raises concern about the contribution of inter-individual variability in the connectivity as a potential confound to the findings. Differences between stimulated and non-stimulated animals would seem to reveal variability comprising both the effect of TUS as well as any across-individual differences. The concern is that the latter can be substantial. Related, in the case of the TUS conditions, fMRI data collection began only after completion of the TUS train.Specifically, our concern arises from the description of the procedures, stating that 'three animals participated in experiments 1 and 2, including the control condition (all males, mean age and weight at time of scan: control, 5.5 years, 10.2 kg; SMA TUS, 5.6 years, 10.7 kg; FPC TUS, 6.1 years, 11.8 kg).' This implies that each animal took part in one condition. Yet it is also stated that the same animals 'took part in experiments 1 and 2', and 'The same three animals participated in experiments 1 and 2, and a control experiment conducted in the absence of TUS.' At the same time, it is stated that 'while experiments 1 and 2 followed a within-subject design, for experiment 3 we conducted a between-subject analysis where the subjects in the two experiments differed in age.' – this would imply that in E1 and E2 a within-subject comparison was done (though it was not clear whether the control comparison was within or between animals). If you could provide a clearer description of the exact procedures and comparisons, this would greatly help with the discussion.Therefore, to which degree could the results arise from significant across-individual differences (as opposed to TUS + across-subject variability)? One observation that may point to this being at least a possibility is that the connectivity of the stimulated animals does not seem to be converging to that of the non-stimulated animals even after 1.5 hours.It would seem that a within-subject comparison of the functional connectivity before, during (if possible) and after TUS is necessary. This procedure could then be repeated for non-stimulated animals to control for the effect of time on connectivity, i.e. compare pre-TUS FC maps for each animal and then post-TUS or post control period maps, for the two target sites. This would address whether the effects that were observed were significantly larger than those that would be expected from inter-animal differences.However, we are aware that repeating the study in full with a within-subject design might be unfeasible, and would therefore discuss a realistic way forward given everyone agreed about the relevance of your work and its likely impact. We would not necessarily expect another large dataset, and a small set of animals may be able to provide reassurance regarding the point above.And there may also be other ways to address the concerns. Would a 'normative' dataset, to which your group might have access to, provide an alternative route, e.g. resting state datasets from other monkey MRI projects? We discussed whether comparison of your animals with such a dataset potentially address the concerns, e.g. by showing how each animal differs from such a normative dataset? As mentioned above, we're happy to hear suggestions given the more substantial expertise on these type of data in your group.There would be other comments should we proceed, but we first wanted to hear your thoughts and how you may be able to address the concerns.

Thank you very much for considering and reviewing our manuscript. We truly appreciate the opportunity to respond to one of the main comments before a binding recommendation can be made.

We understand that a critical sentence concerning this comment is our statement that: "three animals participated in experiments 1 and 2, including the control condition”. We understand that our phrasing might have given the impression that each animal participated in only one condition. In fact, each of the three animals participated in all three of the conditions. If we name the three animals A, B, and C, we could state: “Three animals (A, B, and C) participated in experiment 1. The same three animals (A, B, and C) participated in experiment 2. Lastly, the same three animals (A, B, and C) were scanned following the control conditions against which the results of experiment 1 and 2 are compared. Accordingly, experiments 1 and 2 are within-subject designs with n=3.”

One suggestion arising from the reviews would be to acquire data before and after TUS. In fact, that is exactly what we have done. We first acquired the control condition, after which the animals underwent TUS over SMA (experiment 1), followed by TUS over FPC at least one week later (experiment 2). This also entails that while the effects of TUS did not converge back to baseline within 2 hours, they were not permanent either, as evidenced by the fact that effects observed immediately following TUS over SMA did not carry over to the next session.

In summary, this means that individual differences cannot be mediating apparent effects of interest. For example on the in Figure 2, we compare the effect, on SMA, of TUS to SMA or the control site FPC. The data come from the same individuals but only the stimulated area exhibits the effect of interest. On the right side of Figure 2, however, everything is swapped round and now the only effect of interest is from the FPC when FPC is stimulated. Figure 3 is organized to a similar principle.

Further evidence that effects are not due to individual differences comes from experiment 3) in which we show that we can reproduce the same FPC effects in a second group of animals (let’s call these animals D, E, and F).

In the same light, it was suggested to acquire data not only before and after TUS, as we have done, but perhaps also during TUS, if possible. This is indeed technically feasible. However, placing the TUS transducer and coupling cone close to the head and stimulating while inside the scanner bore introduces MR artefacts. There is also a possibility that the TUS protocol induces audible skull resonance at frequencies caused by low-frequency modulations of the ultrasonic sound wave. To exclude these artefacts completely, we have opted for an offline approach.

For completeness, we would like to highlight that our validation experiment (experiment 3, with TUS over FPC), is indeed a between-subject comparison: three new animals (D, E, and F) underwent TUS over FPC and their results are compared to the control condition of animals A, B, and C. In essence, here the control condition of animals A, B, and C acts as a normative dataset against which D, E, and F are compared.

To conclude, we apologise for any misunderstanding that our wording might have caused. However, we would like to highlight that we have in fact performed a full within-subject comparison, as suggested by the reviews, and that inter-animal differences could therefore not have contributed to the TUS-induced differences that we observed.

[Editors’ notes: the authors’ response after being formally invited to submit a revised submission follows.]

Thanks for the quick reply, which has greatly clarified things. We're happy to invite a revision based on these, and are confident you can provide these within the usual timeframe.

Thank you for asking for this additional information in such a straightforward manner. We hope we have addressed any remaining queries but if there is anything that is still unclear please get in touch. In the responses below we have attempted to address each of the remaining concerns expressed by the reviewers.

In addition to noting some of the points that you raised in your previous letter we also noted in the reviewer comments several points dealing with the pitfalls of undertaking multiple statistical comparisons and a suggestion that this may have occurred when we examined connectivity changes across conditions. Although we reply to these points in much more detail below, it may be worth stating briefly here that we avoid such potential problems by using single indices – cosine similarity metrics – to summarize entire fingerprints of connectivity. It is these single indices that we employ for statistical inference.

Perhaps most critically then would be to clarify the procedures throughout (after all, five people could not confidently work this out), leaving no ambiguity to exactly what was done. The key point of reviewer3 (and related reviewer1 and the two editors) can otherwise be ignored. Elsewhere, I am confident that you will find the requested revisions are straightforward to address.

We would like to thank you for helping us clarify a misunderstanding. To re-iterate, for the main analyses in this study we employed a within-subject design whereby all three animals participated in all conditions. In fact, only in experiment 3, where we perform a validation of experiment 2 in a new set of animals, and in experiment 4, where we perform terminal experiments on the animals, did we employ between-subject comparisons.

To further clarify our study design, experimental timeline, pre-processing pipeline, and analyses strategies we have now included a new figure in the revised manuscript, Figure 1.

We have further clarified our experimental design in the revised manuscript as follows:

Abstract

“In a within-subject design, we observe regionally specific TUS effects for two medial frontal brain regions – supplementary motor area and frontal polar cortex. Independently of these site-specific effects, TUS also induced signal changes in the meningeal compartment.”

Introduction

“Here we focused on the effects of TUS outlasting the stimulation period, investigating the impact of 40 s trains of TUS on measurements of neural activity in three macaque monkeys provided by functional magnetic resonance imaging (fMRI) up to 2 hours after stimulation (Figure 1, top panel). […]We also validated the results of experiment 2 in a new set of three different animals (experiment 3).”

Figure 3 legend. “Coupling of activity between stimulated areas and the rest of the brain in experiments 1 (SMA) and 2 (FPC). The left panels show activity coupling between SMA and the rest of the brain in the control state (a), after SMA TUS (b), and after FPC TUS (c). […] Connectivity seed regions are indicated with black asterisks. Key anatomical features are labelled in panel (a): pos, parieto-occipital sulcus; cal, calcarine sulcus; cgs, cingulate sulcus; ps, principal sulcus; as, arcuate sulcus; cs, central sulcus; ips, intraparietal sulcus; sts, superior temporal sulcus; ls, lunate sulcus.”

Discussion

“In experiments 1 and 2, adopting a within-subject design with three animals, we found that TUS application produced different effects when applied to different brain regions: the SMA, a part of motor association cortex, and FPC a part of granular prefrontal cortex (Figure 3, 4).”

Materials and methods

“Subject details

For this study, six healthy male macaques (Macaca mulatta) were stimulated with transcranial focused ultrasound and scanned to acquire resting state functional magnetic resonance (rs-fMRI) and anatomical MR images. […] In all cases they complied with the European Union guidelines (EU Directive 2010/63/EU).”

Figure 3—figure supplement 1 legend. “Specific patterns of change in the coupling of activity between stimulated areas and the rest of the brain were replicated in experiments 2 and 3. […] All conventions as in Figure 3. Lighter coloured error bands indicate the standard-error of the mean across individual animals.”

Other points you should address:In addition, we'd like you to provide additional detail on the procedures, as raised by reviewer 3.Effect sizes remained a concern (see comments by reviewer 1 and reviewer 3). These may be resolved by revising the figures, but addressing the concern about the procedures for determining and analysing connectivity fingerprints would be helpful. Please also clarify the concern regarding the reporting of the statistical procedures.Some additional detail on exactly how the fingerprints of stimulated animals were compared to non-stimulated animals (i.e., how statistical significance of the fingerprints was computed) would be helpful. For example, the Materials and methods section merely refers to a paper (Mars et al., 2016), which refers to another paper (Nichols and Holmes, 2007) on the important issue of how the permutations were performed.

In the revised manuscript we have attempted to make our analysis approach clearer and we have given particular emphasis to the statistical inference drawn on the connectional fingerprints.

Materials and methods

“Importantly, rather than examining activity coupling between the seed area and each of the fingerprint ROIs in turn and risking potential false positive results, we compared the overall pattern of coupling using the method devised by Mars and colleagues (Mars et al., 2016); non-parametric permutation tests were performed on cosine similarity metrics summarizing pairs of fingerprints (SMA TUS versus control and FPC TUS versus control). […] The relatively small number of TUS sites, animals, and fMRI runs allowed us to exhaustively test all possible permutations (24309) to obtain the true probability of rejecting the null hypothesis.”

In addition, we have clarified the origin of the ROIs used and the rationale for selecting them. The revised text reads as follows:

Materials and methods

“We report on the whole-brain functional coupling of the stimulation sites in the control state and after TUS, adopting a seed-based correlation analysis approach. […] Lastly, to test for specificity of the TUS effects, we included SMA as a target for the FPC fingerprint and FPC as a target for the SMA fingerprint.”

To construct regions-of-interest (ROI) for SMA and FPC circles of 4 mm radius were drawn on the cortical surface around the point closest to the average stimulation coordinate (Figure 2), in both the left and the right hemisphere. The same procedure was used to define other bilateral cortical regions of interest, based on literature coordinates (Mars et al., 2011; Neubert et al., 2015, 2014; Sallet et al., 2013), to serve as seeds for connectivity analyses (A1, Figure 4G-I; POp, Figure 6, Figure 6—figure supplement 1) or targets for the fingerprint analyses (Figure 4). Apart from the stimulation sites and primary auditory cortex, all ROIs were selected based on their known anatomical connectivity, of relevance because they are known to be distinctively strongly or distinctively weakly connected with the stimulation sites.

For some analyses multiple tests were conducted. What procedures for corrections for multiple comparisons were in place throughout?

In the revised manuscript we put additional emphasis on the manner in which our statistical approach avoids the potential pitfalls associated with making multiple comparisons. The revised text reads as follows:

Materials and methods

“To report on the full extent of the TUS effects and the simple effects that drive any observed differences we reproduce the whole-brain functional connectivity maps unthresholded for the control, SMA TUS, and FPC TUS conditions separately. […]Importantly, rather than examining activity coupling between the seed area and each of the fingerprint ROIs in turn and risking potential false positive results, we compared the overall pattern of coupling using the method devised by Mars and colleagues (Mars et al., 2016); non-parametric permutation tests were performed on cosine similarity metrics summarizing pairs of fingerprints (SMA TUS versus control and FPC TUS versus control).”

Again we note, as we have noted above, how the precise nature of the approach used precludes multiple comparisons. The cosine metric provides a single index summarizing each fingerprint. Comparisons between fingerprints are not based on multiple comparisons of each point on each fingerprint but instead by a single comparison between the cosine metrics. We note the relevant section of the manuscript again below:

“Importantly, rather than examining activity coupling between the seed area and each of the fingerprint ROIs in turn and risking potential false positive results, we compared the overall pattern of coupling using the method devised by Mars and colleagues (Mars et al., 2016); non-parametric permutation tests were performed on cosine similarity metrics summarizing pairs of fingerprints (SMA TUS versus control and FPC TUS versus control). Each connectional fingerprint can be represented as a multidimensional vector, with the number of dimensions corresponding to the number of target ROIs in the fingerprint. The cosine similarity metric quantifies the angle between two of such multi-dimensional vectors. As such, it takes into account the shape of the fingerprint irrespective of its mean amplitude and results in a single metric per pair of fingerprints, negating the necessity for correcting for multiple comparisons across fingerprint targets.”

The sizes of the correlation vectors being compared do not seem to be provided either in the text or figure captions, but would be useful to see.

In the revised manuscript we now specify the exact length of the fMRI timeseries (correlation vectors) in volumes.

Materials and methods

“To represent subject effects, the timeseries from the three runs were concatenated to create a single timeseries of 3×800-5=2385 volumes per animal per intervention (control, SMA TUS, FPC TUS). To represent group effects the run-concatenated timeseries of all animals were combined using a group-PCA approach resulting in a series of 200 volumes representing the principal components (Smith et al., 2014).”

The choice of frontal polar cortex seems valid as a control because it has a very distinct functional connectivity from SMA. Is the SNR comparable for these two regions? To rule out the possibility that differences between FPC and SMA are simply related to SNR differences, could you should show temporal SNR maps and quantify these differences, if any?

The temporal signal-to-noise of the BOLD signal measured from a given brain area could impact the estimated coupling strength (Pearson’s r) of that area with other brain regions. This is an undesirable and common confound in functional connectivity analyses. Nonetheless, main differences in tSNR, either between brain areas or between control and TUS, cannot straightforwardly drive the TUS effects observed in our study. Namely, the observed effects are not only specific to the stimulation condition (control vs. TUS), but also to the site (SMA or FPC). As such, any potential effect on tSNR would have to exhibit an interaction effect of TUS and site. For example, we don’t just compare the impact of SMA TUS on SMA’s connectivity with FPC TUS’s impact on FPC’s connectivity; we also compare SMA TUS’s impact on SMA connectivity with FPC TUS’s impact on SMA connectivity and examine SMA connectivity at baseline. In the case of FPC, we compare FPC TUS’s impact on FPC connectivity with SMA TUS’s impact on FPC connectivity and FPC connectivity at baseline.

Moreover, even in the hypothetical case that such an interaction were present, this would result in global change of the correlation values estimated for each stimulation site. It would not be able to explain how after TUS the coupling of a seed region might be enhanced with some areas but disrupted with others.

Accordingly, to exclude the possibility of a three-way interaction of stimulation condition x site x other-brain-areas we quantified the temporal variability of the BOLD signal at every point in the brain, including the SMA and FPC, in all three TUS conditions (control, SMA, FPC). Please note that the mean offset of the signal is largely uninformative in BOLD fMRI and detrimental for correlational analyses. We have therefore quantified the temporal standard deviation of the signal instead. This reveals a pattern of variation across the brain that reflects basic properties of the data acquisition and signal (e.g. proximity to the RF receive coils) but importantly this pattern is not modulated by TUS and is not meaningfully different between SMA and FPC regions.

We have included these new analyses in the revised manuscript in the Results section and as a figure supplement:

Results

“While BOLD fMRI cannot provide an absolute measure of neural activity, we can characterize how homogeneous the activation signal is within the stimulated region, as quantified by the coupling strength of the signal at each point in the stimulated region of interest to all other points in that region. […]This suggests that TUS leaves intact basic haemodynamics and neurophysiology and instead has a circumscribed and specific impact on the coupling of the stimulated region with the rest of the brain.”

It is virtually impossible to appreciate the differences in the FC maps in Figure 2. The addition of difference maps might help here (e.g. TUA over SMA minus control).

In the revised manuscript we have now included an additional figure (Figure 5) showing the difference maps for SMA TUS minus control and for FPC TUS minus control. In this figure we have also drawn our ROIs as used in the fingerprint analyses. This figure shows the direction and extent of the TUS induced changes to whole-brain coupling and highlights that the effects were neither limited to nor maximal in the ROIs selected for the fingerprint analyses.

The revised text reads as follows:

Results

“Following ultrasound stimulation, SMA changed its coupling with the sensorimotor system, anterior and posterior cingulate, anterior temporal, inferior parietal, and prefrontal cortex (Figure 3B). […]This pattern not only emerges from the fingerprint analyses, but constitutes a principle evident across the brain, as illustrated by whole-brain differential SMA-connectivity maps of the effect of SMA TUS (Figure 5).”

Experiment 2, TUS modulation of FPC connectivity

[…]

“These results are apparent in the whole brain functional connectivity maps for the FPC region (Figure 3, compare panels D and F, representative changes highlighted by dashed black circles) and on the whole brain differential connectivity maps in Figure 5 (panels D and F).”

Regarding the acoustic parameter, 30 ms burst in every 100 ms (i.e. pulse repetition of 10 Hz; i.e. duty cycle of 30%), seems unconventional, even though they appear to be effective here. 10 Hz is well below the audible frequency, which helped to disregard effects from auditory involvement. However, an additional rationale for the choice of parameter (or brief summary of the literature) will be helpful.

The sonication protocol we employed in this study was based on a preliminary work on sustained ultrasonic neuromodulation (Ahnine et al., 2018). This protocol was similar to the protocol used by Dallapiazza and colleagues when observing short-lived offline effects following ultrasound targeted at swine thalamus (Dallapiazza et al., 2017). This protocol is characterised by a repetitive structure with a relatively high rate of acoustic energy deposition. Previous work has suggested that ultrasound protocols with a higher acoustic intensity (Isppa > 5 W/cm2) and a higher duty cycle (> 5%) might be most likely to elicit neuromodulation outlasting the sonication period (Kim et al., 2015; Yoo et al., 2011). We have taken this into consideration when choosing our parameters, while aiming to maintain safe stimulation intensities that do not lead to excessive thermal modulation or microstructural changes.

We have now included these considerations in the revised text:

Introduction

“To date ultrasonic applications are primarily focused on direct ‘online’ effects. […] We have built on such protocols when designing the current experiment.”

With regard to the acoustic and Thermal modelling, the conservative approach taken here may over-estimate the actual temperature rise in the skull and the brain. Although additional experiments are not necessary, realistic estimation of temperature rise using phantoms would be desirable. If access to such data exist we would encourage you to include these.

In this study the thermal modelling serves to provide an informed estimate whether the employed sonication protocol might have caused thermal damage and whether thermal effects could potentially underlie the observed neuromodulatory effects. We feel that for these purposes a rigorous and perhaps somewhat conservative approach might be most appropriate. While at the time of the study we did not have access to realistic phantoms of macaque heads we agree that this would be a worthwhile endeavour to further characterise the current protocol.

We have revised the text as follows:

Discussion

“Similarly, the thermal modelling approach we adopted here was designed to estimate an informed upper-bound on potential thermal effects, but when further developing this protocol additional simulation validations – for example based on phantom measurements – might be informative.”

In the histological analysis, a rather limited staining approach is used (i.e. Cresyl Violet and H&E) to evaluate the sonicated brain tissue. Although not absolutely needed in present study, there are other types of histological analyses to assess the presence of tissue damages (such as apoptotic activities or ischemic damages). At the very least, please reflect on these limitations in the Discussion.

Our histological analyses were based on previous publications that employed NISSL H&E staining to identify microhaemorrhages following either high-intensity or repetitive low-intensity ultrasound stimulation (Dallapiazza et al., 2017; Lee et al., 2016b). By conducting a similar analysis, we were able to show that no similar tissue damage was associated with our 40s stimulation protocol. To further characterize potential confounding effects, we inspected T1w images for possible sign of oedema. T1w images were collected immediately after the 3 runs of resting-state data used for the main analysis of this paper. We did not observe any sign of oedema 2h post-stimulation.

Finally, we agree with the reviewer that future histological analyses will need to be conducted to further characterized the impact of TUS on the brain (see for example Tufail et al., 2010).

We have amended the revised text as follows:

Results

“Experiment 4, meso- and micro-structural analyses

Some higher intensity ultrasound stimulation protocols, distinct from those used here, have been shown to induce thermal lesions or haemorrhage following cavitation (Elias et al., 2013). […]Neither were any signs of neuronal alteration or haemorrhage observed in histological analyses of three macaques following pre-SMA TUS (Figure 8).”

Discussion

“Some caution might also be warranted in relation to the potential of ultrasound to cause microstructural damage, especially when stimulating at higher intensities, longer durations, or for more repetitions. While our structural MRI and histological analyses did not reveal any evidence of damage and were comparable to previous studies (Dallapiazza et al., 2017; Lee et al., 2016b), it would be of interest in the future to include additional histological indices of apoptotic or inflammatory processes (Tufail et al., 2010) to further assess the safety of TUS.”

Materials and methods

“Histology

In experiment 4, prior to histological examination, animals were anaesthetized with sodium pentorbarbitone and perfused with 90% saline and 10% formalin. […] Every tenth section, two adjacent sections were retained for analysis whereby one was stained with Cresyl Violet (Nissl body staining) and the other with Haemotoxylin and Eosin (H&E staining), in line with previous studies (Dallapiazza et al., 2017; Lee et al., 2016b).”

Transcranial focused ultrasound was abridged to 'TUS'. As the method utilized the focused version (i.e. FUS), the change in acronym to tFUS might be more intuitive.

We acknowledge that there are indeed several acronyms currently in use in the literature, all referring to the same approach. We feel that in such circumstances it would be helpful to aim for consistency. In one way, one could argue that it would be consistent to adopt the ‘tFUS’ acronym used by several US-based labs. In another way, one could argue that it would be consistent to adopt the ‘TUS’ acronym we have used in previous publications employing the same technique.

We have expressed a preference for the ‘TUS’ acronym, here and previously in our work. Namely, this acronym avoids any redundant or implicit aspects of the technique: virtually all reversible ultrasound stimulation protocols to date use focused, pulsed, low-intensity, low-frequency ultrasound. At the same time, this acronym maintains its reference to the core features: it passes through the skull (signifying it is targeted at the brain and is therefore minimally invasive), it utilises ultrasound to deliver energy, and it is a form of stimulation (rather than imaging). In this way the acronym ‘TUS’ also provides an easily recognizable complement to other non-invasive brain stimulation approaches: TMS, TCS, and TES.

Inclusion of some relevant published work may be helpful, at your discretion: (1) fMRI-tFUS human study(ies) and (2) recent findings regarding the presence of long-term neuromodulatory effects of tFUS in rodents.

In the revised manuscript we have tried to give a more thorough overview of the relevant literature. In addition, we now also refer to recent review publications that provide a more comprehensive overview of the field than we are able to do in this original research report. We have adapted the manuscript as follows:

Introduction

“Here we report on a particular protocol of low intensity pulsed transcranial focused ultrasound stimulation (TUS) that we show induces a sustained period of neuromodulation in primates without inducing structural damage. […] We have built on such protocols when designing the current experiment.”

Reviewer 1:Verhagen and colleague used resting-state fMRI in macaque monkeys to investigate the effects of transcranial focused ultrasound (FUS) on functional brain connectivity. The study appears to be carefully conducted and it reveals some important aspects of FUS. First it shows that FUS increases local functional connectivity and mainly decreases distant functional connectivity. Second it demonstrates that the effects last for at least 2 hours. Overall the effects appear to quite small but they seem to be consistent across animals (Fg. 3c). I have a few major points that need to be addressed.1) I assume the authors picked frontal polar cortex because it has a very distinct functional connectivity from SMA. Based on my experience, the SNR can be quite low for this anterior area. To rule out the possibility that differences between FPC and SMA are simply related to SNR differences, the authors should show temporal SNR maps.

Please see above responses.

2) I found it virtually impossible to see any differences in the FC maps in Figure 2. The addition of difference maps might help here (e.g. TUA over SMA minus control).

Please see above responses.

3) It is not clear from the paper how often each animal was actually scanned. Did the authors only perform one SMA and one FPC stimulation session per monkey or did they conduct multiple sessions?We performed one session per monkey per condition. In total we performed three sessions per animal across experiments 1 and 2, and one session per animal in experiment 3. Each session consisted of three consecutive runs. This is further clarified in the new Figure 1 and in the revised text:

Introduction

“In a within-subject design, the same three animals participated in experiments 1 and 2, and a control experiment conducted in the absence of TUS (Figure 1). As such, fMRI was acquired in three conditions for all animals: following SMA TUS, following FPC TUS, and in a control state. In turn, each of the MRI sessions consisted of three consecutive runs.”

Materials and methods

*“*Macaque MRI acquisition

We acquired one MRI session per monkey per condition: in total we performed three sessions per animal across experiments 1 and 2, and one session per animal in experiment 3.”

Reviewer 2:[…]Here are a few suggestions, with moderate importance, which may improve themanuscript.1. Regarding the acoustic parameter, 30 ms burst in every 100 ms (i.e. pulse repetition of 10 Hz; i.e. duty cycle of 30%), is bit unconventional, but appeared to be effective. 10 Hz is well below the audible frequency, which helped the authors to disregard effects from auditory involvement. Additional rationale for the choice of parameter or brief review on literatures on the parameters in the Discussion section will be helpful.

Please see above responses.

2. In regard to the acoustic and Thermal modeling, the study seems to take conservative approach, which may over-estimate the actual temperature rise in the skull and the brain. Although additional experiments are not necessary, realistic estimation of temperature rise using phantoms will help to ascertain the non-thermal effects of tFUS.

Please see above responses.

3. In histological analysis, authors performed a rather limited staining (i.e. Cresyl Violet and H&E) to evaluate the sonicated brain tissue. Although not absolutely needed in present study, there are other types of histological analyses to assess the presence of tissue damages (such as apoptotic activities or ischemic damages), and authors are advised to reflect these limitations in the Discussion.

Please see above responses.

4. Transcranial focused ultrasound's acronym was defined as 'TUS'. As the method utilized the focused version (i.e. FUS), the change in acronym to tFUS is strongly suggested.

Please see above responses.

5. Although the detailed references are given in the technical methods section,surprisingly, only a few previous investigations on the matter of main subjects (fMRI and long-term effects of tFUS) are given. Inclusion of (1) fMRI-tFUS study(ies) among humans and (2) recent findings regarding the presence of long-term neuromodulatory effects of tFUS on rodents (a simple Pubmed search will do) is advised for general audience.

Please see above responses.

Reviewer 3:[…]I believe that a more suitable approach would have been to compare functional connectivity before, during (if possible) and after TUS within an animal. This procedure would then be repeated for non-stimulated animals to control for the effect of time on connectivity. It would also have been more convincing to observe effects when comparing the entire functional connectivity matrices (instead of just the pruned fingerprints, especially since little information was provided as to how the pruning was achieved).

We hope we have now clarified that we have performed within-subject comparisons in multiple animals unconfounded by effects of order and time. This constitutes an even more powerful experimental design than the one proposed by the reviewer. Indeed, a comparison between animals of a change in functional connectivity over time, contrasting TUS animals with control animals, would require one to test a mixed-level interaction effect with somewhat reduced statistical power. Similarly, a statistical comparison of whole-brain functional connectivity would likely suffer from either a distractingly high false-positive rate, or, when stringently correcting for multiple comparisons, from a detrimentally high false-negative rate. The fingerprint analysis is designed specifically to avoid these pitfalls, allowing one to leverage a-priori knowledge of neuroanatomy.